# Dicer promotes genome stability via the bromodomain transcriptional co-activator BRD4

M. J. Gutbrod [1,2], B. Roche[2], J. I. Steinberg[2,3], A. A. Lakhani[1,2], K. Chang[2], A. J. Schorn[2] &
R. A. Martienssen [2,4 ✉]

RNA interference is required for post-transcriptional silencing, but also has additional roles in transcriptional silencing of centromeres and genome stability. However, these roles have been controversial in mammals. Strikingly, we found that Dicer-deficient embryonic stem cells have strong proliferation and chromosome segregation defects as well as increased transcription of centromeric satellite repeats, which triggers the interferon response. We conducted a CRISPR-Cas9 genetic screen to restore viability and identified transcriptional activators, histone H3K9 methyltransferases, and chromosome segregation factors as suppressors, resembling Dicer suppressors identified in independent screens in fission yeast. The strongest suppressors were mutations in the transcriptional co-activator *Brd4*, which reversed the strand-specific transcription of major satellite repeats suppressing the interferon response, and in the histone acetyltransferase *Elp3*. We show that identical mutations in the second bromodomain of *Brd4* rescue Dicer-dependent silencing and chromosome segregation defects in both mammalian cells and fission yeast. This remarkable conservation demonstrates that RNA interference has an ancient role in transcriptional silencing and in particular of satellite repeats, which is essential for cell cycle progression and proper chromosome segregation. Our results have pharmacological implications for cancer and autoimmune diseases characterized by unregulated transcription of satellite repeats.

[1] Cold Spring Harbor Laboratory School of Biological Sciences, Cold Spring Harbor Laboratory, Cold Spring Harbor, NY, USA. [2] Cold Spring Harbor Laboratory, Cold Spring Harbor, NY, USA. [3] Medical Scientist Training Program (MSTP), Renaissance School of Medicine at Stony Brook University, Stony Brook, NY, USA. [4] Howard Hughes Medical Institute, Cold Spring Harbor Laboratory, Cold Spring Harbor, NY, USA. ✉email: martiens@cshl.edu

Canonical RNA interference (RNAi) silences genes and transposable elements (TEs) through microRNAs[1], small interfering RNAs (siRNAs), or PIWI-interacting RNAs (piRNAs)[2]. However, RNAi also controls genome instability via chromosome dosage and segregation[3–7], transcription termination, and the DNA damage response[8–13]. For example, the fission yeast Schizosaccharomyces pombe does not have microRNAs, but RNAi plays an important role in heterochromatic silencing and chromosome segregation through a process called co-transcriptional gene silencing (CTGS)[14,15]. Chromosome segregation defects in RNAi mutants are associated with a reduction in histone H3K9 methylation at the centromere[16,17], increased transcription, and the loss of cohesin at the locus. Additionally, RNAi becomes essential in the cell divisions preceding quiescence[18]. Dicer (Dcr1) and CTGS remove RNA polymerases that would otherwise transcribe centromeric repeats, preventing collision with replication and DNA damage[12,13]. Polymerase removal is also required for long term survival in quiescence[18].

Similar mitotic chromosomal defects result from perturbing Dicer in other eukaryotes[19–22] including human[21,23] and mouse cells[24]. While the deletion of the microRNA-specific factor Dgcr8 in mouse embryonic stem cells (mESCs) generates only mild cell cycle stalling in G1[25], the deletion of Dicer1 in mESCs generates severe proliferation defects, strong stalling in G1, a significant increase in apoptosis, and an accumulation of transcripts from the major (pericentromeric) and minor (centromeric) satellites, all of which must be microRNA-independent[26–28]. In mouse, DICER1 associates with major satellite RNA and pericentromeric chromatin[29], which is characterized by H3K9me2/3 and HP1 proteins as in fission yeast[30,31]. H3K27me3 is also present and is redundant with H3K9me2/3[32,33]. In Dicer1−/− mESCs, widely differing phenotypes have been reported[26–28], and one explanation might be the accumulation of mutations that allow stalled Dicer1−/− cells to proliferate[28]. While changes in DNA methylation were initially hypothesized to be partially responsible for proliferation defects[34,35], follow-up studies have demonstrated little change in DNA methylation levels in Dicer1−/− mESCs which do not explain the phenotypes observed[36]. Genetic suppressors arise in Dicer mutants of fission yeast when they exit the cell cycle and as RNAi becomes essential, resulting in the selection and outgrowth of suppressed strains[18]. We hypothesized that when Dicer1−/− mESCs stall in G1, an essential function is revealed that needs to be suppressed in a similar way. We therefore performed a CRISPR-Cas9 genetic screen in Dicer1−/− mESCs, focused on chromatin modifiers found in similar screens of S. pombe[18] to identify genetic suppressors and characterize the molecular mechanism of the Dicer1−/− viability defects.

lines (Fig. 1C), in addition to aneuploidies commonly found in mESC, that might contribute to survival[38,39]. Transcriptome analysis of these lines (Fig. 1D) also agreed with previous studies, including increased expression of apoptosis-promoting genes (Fig. S3A), but the most upregulated non-coding transcripts were from pericentromeric satellite repeats and endogenous retroviruses (ERVs) (Fig. 1D and Supplementary Data 1), potentially accounting for the strong interferon response (Fig. 1E)[26,40]. Transcriptomes from Dicer1−/− clonal lines differed substantially from that of freshly induced cells, consistent with strong selection for viable clones (Fig. S3B). Intriguingly, we observed a reverse strand bias in major satellite transcripts in Dicer1−/− mESCs that switched to the forward strand after selection for viable clones (Fig. 1F). No such switches were observed in other genes or TEs genome wide.

S. pombe and the nematode C. elegans have RNA-dependent RNA Polymerases (RdRP) that generate double stranded templates for Dicer. Resulting small RNAs from transposons and repeats then guide histone H3K9 methylation[15,41]. Mammals lack RdRP, and we did not detect DICER1-dependent small RNAs from the major satellite transcripts in wild-type mESC (Fig. S3C), although we did detect degradation products in Dicer1−/− cells resembling primary RNAs (priRNAs) observed in S. pombe[42]. DICER1-dependent siRNAs from ERVs were limited to miRNA, while 3'tRNA fragments that match ERVs accumulated to higher levels in Dicer1−/− cells[43] (Fig. S3C). The lack of DICER1-dependent siRNA is consistent with the absence of an RdRP. Furthermore, the knockdown of all mouse Argonaute proteins simultaneously did not affect the proliferation of Dicer1−/− mESCs in a viability assay (Fig. S3D–G), also demonstrating that small RNA are unlikely to be the primary driver of the Dicer1−/− phenotypes. We also performed ChIP-seq, and observed a modest reduction in H3K9me3 at some ERVs, namely IAP and ETn, as well as at LINE1 transposable element loci (Figs. S4A–D, and S5B), as described previously[44], which might be related to the loss of microRNA. In contrast, we found a modest increase in H3K9me3 at the major satellite (Fig. S4C), which was confirmed by immunofluorescence (Fig. S5C and D) as well as increased HP1β binding (Fig. S5E). We found little change in CENPA at the minor satellite (Fig. S5F), but we did detect a 2-fold decrease of H3K27me3 at the major satellite and a reduction of the H3K27 methyltransferase EZH2 at these loci (Fig. S5G and H). While it is possible that the increase in H3K9me3 at the pericentromeres was guided by priRNA[42], we conclude that loss of small RNA and H3K9me3 could not account for the strong phenotypes observed in Dicer1−/− mESCs. We therefore turned to a more unbiased strategy to determine the underlying mechanism.

## Results

### Dicer1 is essential for proliferation and chromosome segregation in mouse ES cells.
We reproduced the severe phenotype of Dicer1−/− mESCs by using inducible homozygous deletion of the RNase III domains upon treatment with hydroxytamoxifen (OHT) at day 0 (d0)[28,37]. We also derived stable clonal lines from rare single mutant cells that survived induction (Dicer1−/− clones), but only after several weeks, indicating selection for suppressors had occurred (Fig. S1A). During the induction timecourse, we observed strong proliferation defects, accumulation of cells in G1, and increased cell death and DNA damage (Fig. S1B,C), as previously reported[26,28]. We also observed frequent chromosome lagging, bridging, and micronuclei indicating segregation defects (Figs. 1A, B, S2A and B). No such phenotypes were observed in a Dgcr8 knockout cell line (Figs. 1B and S2B), ruling out microRNA-based mechanisms[23]. Whole genome sequencing revealed partial trisomy in two Dicer1−/− clonal cell

### Brd4 and Elp3 mutations suppress the Dicer1−/− phenotype.
We performed a CRISPR-Cas9 screen for genetic modifiers that rescued the Dicer1−/− proliferation and viability defects. In order to avoid aneuploids and second-site suppressors in established clonal cell lines, we introduced a domain-focused single guide RNA (sgRNA) library into wild-type cells and then induced homozygous Dicer1−/− deletion with OHT treatment (Fig. S6A). The library targeted 176 genes responsible for chromatin modification, and sequencing at d12, d16, and d20 revealed guides that were either enriched (genetic suppressors) or depleted (genetic enhancers) after selection (Supplementary Data 2). Of the suppressors, the bromodomain transcription factor Brd4 and the histone acetyltransferase Elp3 were outliers as the strongest hits at all three timepoints (Fig. 2A). Additionally, mutations in the H3K9 methyltransferases Ehmt2 (G9a), Ehmt1 (GLP), Suv39h1, and Suv39h2 were ranked as genetic suppressors while the H3K27 methyltransferase Ezh2 was a strong enhancer

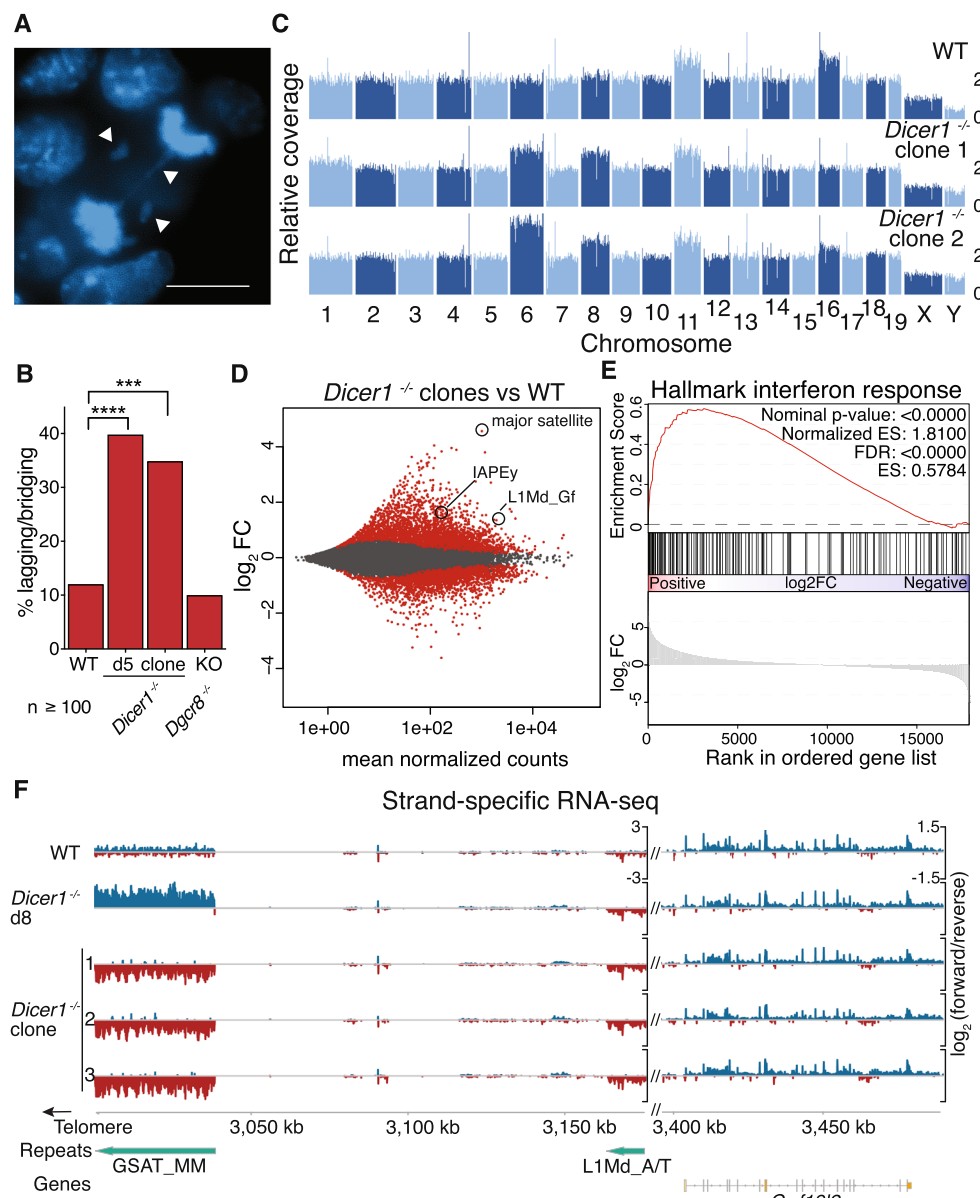

**Fig. 1 *Dicer1* promotes proliferation and chromosome segregation in mESCs and regulates the major satellite transcript in a strand-specific manner.**
**A** DAPI staining of lagging and bridging chromosomes (arrows) in *Dicer1*−/− anaphase cells. Scale bar: 10 μm (*n* = 100 cells). **B** Quantification of lagging/bridging chromosomes (**** − *p* < 0.0001, *** − 0.001 > *p* > 0.0001, - Fisher's exact test, two-tailed). Chromosome mis-segregation is prevalent in *Dicer1*−/− cells 5 days after deletion of the *Dicer1* gene (d5) as well as in clones selected for viability after several weeks (clones), but not in cells deficient for the miRNA processor *Dgcr8*. **C** Genome browser of normalized coverage of whole genome sequencing reads reveals enhanced aneuploidy in *Dicer1*−/− clones. Mouse autosomes are numbered 1-19. Scale bar represents inferred ploidy (median = 2n). **D** Differential expression analysis of *Dicer1*−/− clones compared to uninduced wild type cells by RNA-seq. The major satellite repeat is strongly upregulated. Other non-coding transcripts include endogenous retroviruses (IAP) and LINE elements (L1). **E** Hallmark Gene Sets Enrichment Analysis (GSEA) reveals an activated interferon response in *Dicer1*−/− clones. **F** Strand-specific transcriptional upregulation of a 38 kb major satellite repeat (GSAT_MM) on chr9, but not of nearby L1 TEs or genes. The log₂ normalized ratio of forward (blue) and reverse (red) strand is plotted. One representative replicate of 3 is plotted for WT and *Dicer1*−/− cells 8 days after induction (d8). 3 independent *Dicer1*−/− clones have reversed transcription at the major satellite after selection for viability.

(Fig. S6B). These results were consistent with increased H3K9me3 and reduced H3K27me3 at the pericentromeric satellite in *Dicer1*−/− cells, but as *Brd4* and *Elp3* were clearly the strongest suppressors they were investigated further.

BRD4 was first described as a protein that remains bound to mitotic chromosomes[45], but has since been found to globally regulate Pol II transcription at enhancer and promoter elements[46,47]. BRD4 contains tandem bromodomains (BDs) that recognize acetylated lysines as well as C-terminal domains that activate transcription (Fig. 2E). However, in HeLa cells BRD4 is recruited to pericentromeric heterochromatin under heat stress[48], while in *S. pombe*, the *Brd4* homolog Bdf2 regulates the boundaries of pericentromeric heterochromatin[49]. ELP3 is a histone acetyl-transferase (HAT) and the catalytic subunit of the Elongator complex, which also promotes Pol II transcription through chromatin[50] by acetylating histone H3 and H4 both in the budding yeast *Saccharomyces cerevisiae*[51,52] and in human cell lines[50]. ELP3 is more thoroughly characterized as a tRNA acetyl transferase, which utilizes a radical S-adenosyl-L-methionine (SAM) domain, to modify U₃₄ of some tRNAs[53] (Fig. 2E).

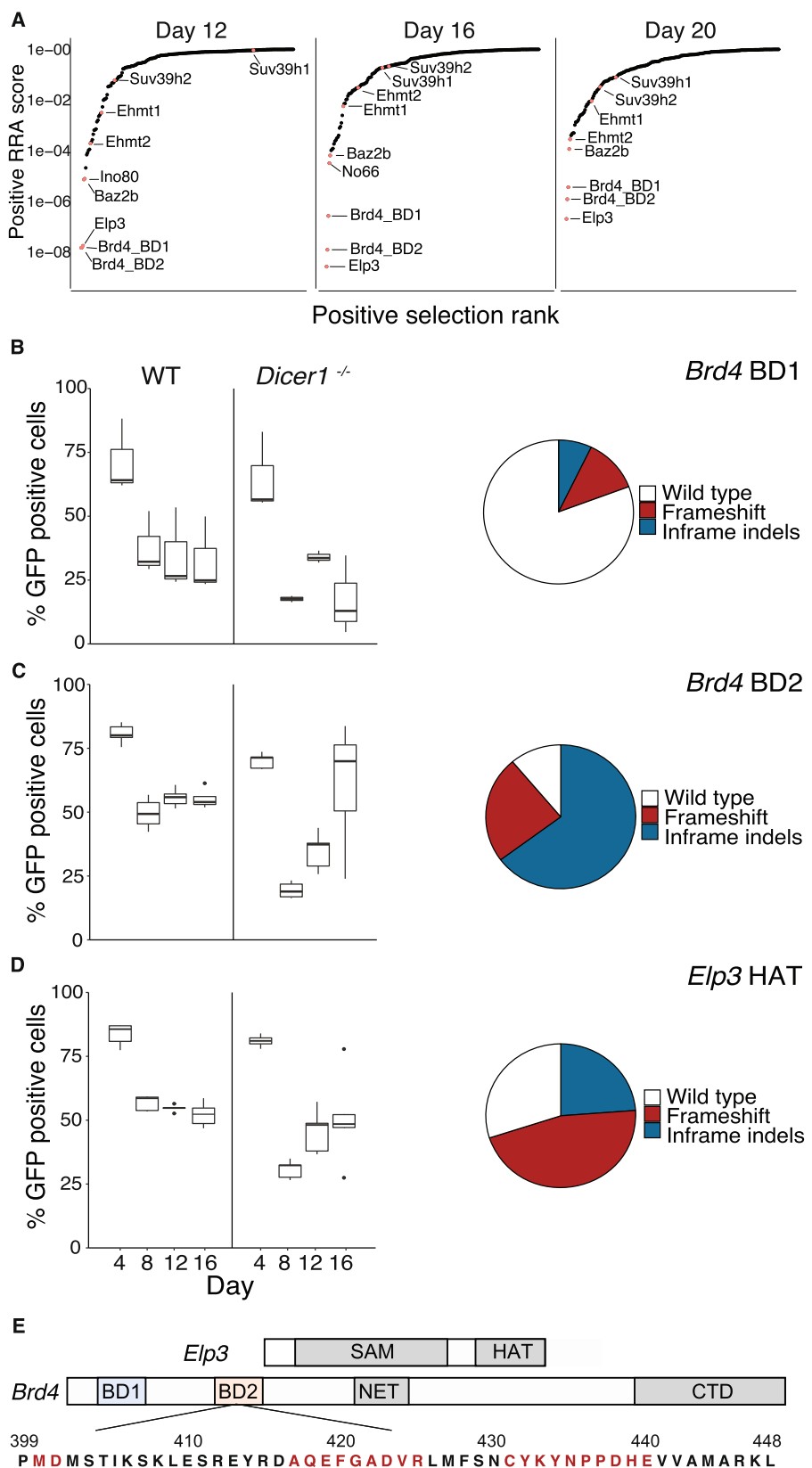

We validated these observations by introducing *Brd4* and *Elp3* individual sgRNAs in parallel together with a green fluorescent protein (GFP) reporter gene. We observed enrichment of GFP positive cells following induction of *Dicer1* perturbation consistent with mutations in these genes being suppressors (Fig. 2B–D). We found that targeting the second bromodomain

(BD2) of *Brd4* was more effective than targeting the first (BD1). At the conclusion of the *Dicer1*$^{-/-}$ timecourse, the majority of BD1 alleles in GFP positive cells yielded a wild type amino acid sequence (Fig. 2B) suggesting a negative selection of deleterious alleles over time, while the majority of BD2 alleles were small in-frame deletions (Fig. 2C). This was consistent with the essential

**Fig. 2 CRISPR/Cas9 suppressor screen in *Dicer1*$^{-/-}$ cells rescues viability by in-frame mutations in *Brd4* and heterozygous frameshifts in the HAT domain *Elp3*. A** A library of small guide RNAs (sgRNAs) targeting chromatin regulators was introduced into *Dicer1*$^{-/-}$ cells and sequenced at three timepoints after tamoxifen-induced mutation of *Dicer1* at day 0 (d0). Individual target genes (labeled) were ranked according to positive RRA score, which ranks genes based on the positive selection of all sgRNAs targeting them. **B**–**D** Individual sgRNAs targeting *Brd4* and *Elp3* were introduced with GFP reporter genes into *Dicer1*$^{-/-}$ cells, and GFP positive cells were counted at the same three timepoints after tamoxifen-induced deletion of *Dicer1* (upper panels). The inferred proportion of sgRNAs targeting the first (BD1) **B** and second (BD2) **C** bromodomains of *Brd4* and the histone acetyltransferase domain of *Elp3* (HAT) **D** are shown as boxplots (center line=median; box=Q1-Q3; whiskers=1.5*IQR deviation from quartiles). The relative proportion of in-frame small deletions and frameshift mutations recovered in each case are plotted as pie charts. *Brd4* BD2 mutations were strongly selected during growth, and were largely in frame. N = 3 independent biological replicates for BD1, N = 5 for BD2 and ELP3; Source Data are provided in the Source Data file. **E** The domain structure of ELP3 and BRD4 is shown as well as the amino acid sequence of the second bromodomain of *Brd4*. Amino acids missing in the small in-frame deletion alleles of *Brd4* BD2 generated by CRISPR mutagenesis are highlighted (red).

function of BD1[54] and suggested that viable BD2 alleles rescued growth. The amino acids frequently deleted in BD2 are highly conserved and important in BRD4 function (Fig. 2E). In contrast, frameshifts were predominant in *Elp3* suggesting heterozygous loss of the HAT domain rescued growth, while maintaining the essential N-terminal SAM domain (Fig. 2D). We examined RNA and protein levels of BRD4 and ELP3 in *Dicer1*$^{-/-}$ mESCs and observed little or no change (Fig. S6C and D) suggesting neither was a target of microRNAs.

We then generated clonal single and double mutant lines using the same sgRNAs. We recovered *Brd4* mutants with heteroallelic indels in BD2 and *Elp3* mutants with heterozygous frameshift mutations in the HAT domain. These lines suppressed the viability defects of induced *Dicer1*$^{-/-}$ mESCs in a luciferase-based metabolism (MT) viability assay (Fig. S6E and F), performing better than induced cells due to clonal selection (Fig. 3A and B). Additionally, the defects were rescued by *Brd4* siRNAs (Fig. 3E), though siRNAs targeting *Elp3* did not suppress the proliferation defect (Fig. 3E), presumably because they also reduced the function of the essential SAM domain. Strikingly, inhibiting Pol II elongation with low concentrations of α-amanitin also rescued viability (Fig. 3C), while this inhibition had no effect on wild type cells (Fig. S7A). Most significantly, the *Dicer1*$^{-/-}$ viability defect was strongly rescued by the small molecule inhibitor JQ1, which specifically inhibits BRD4 and its paralogs (Fig. 3D). While targeting BRD4 with siRNAs or JQ1 does inhibit both bromodomains, we did not observe the deleterious effects of targeting BD1 with these modalities in either *Dicer1*$^{-/-}$ or wild type mESCs (Figs. 3D, E, S7B, and C). This is likely due to the transitory and less efficient nature of these treatments in comparison to generating frameshift mutations with CRISPR as in Fig. 2.

***Dicer1*$^{-/-}$ chromosomal defects are associated with major satellite transcription**. In order to investigate the mechanism of suppression, we performed BRD4 ChIP-seq in wild type and *Dicer1*$^{-/-}$ mESCs. We found a two-fold reduction of BRD4 occupancy at the vast majority of genes (Fig. S8A) making them unlikely targets for rescue by *Brd4* loss-of-function mutations. Intriguingly, one feature with much higher levels of BRD4 was a 38 kb portion of the major satellite repeat found in the mm10 reference genome sequence at the end of chromosome 9 (Fig. 4A), which matched 20 of the top 40 significantly differential BRD4 peaks in *Dicer1*$^{-/-}$ clones (Supplementary Data 3). Next, we determined whether transcript levels were significantly altered upon *Brd4*$^{BD2-/-}$ or *Elp3*$^{HAT+/-}$ mutation in *Dicer1*$^{-/-}$ clones. *Brd4*$^{BD2-/-}$ or *Elp3*$^{HAT+/-}$ mutations alone had only mild effects on transcription (100–200 mostly down-regulated genes), but shared a significant number of targets (28), including known BRD4 targets *Lefty1* and *Lefty2*[55]. Further, the mutation of *Brd4* or *Elp3* in combination with *Dicer1* resulted in differential expression of many transcripts relative to *Dicer1* single mutants

(Fig. S8E), more than half of which were shared. Intersection of the datasets revealed that more than half of the differentially expressed genes had a BRD4 peak immediately upstream (<10 kb) (Fig. S8F). 97 genes were upregulated upon *Dicer1* mutation, downregulated upon *Brd4* and *Elp3* mutation, and were located near a BRD4 ChIP-seq peak, suggesting they were direct targets of both DICER1 and BRD4. We closely examined all 97 of these candidates through – (i). Cross-referencing *Dgcr8* and *Drosha* knockout mESC RNA-seq datasets to eliminate microRNA target genes, (ii). Searching for homologs in *S. pombe* as Dicer-dependent centromeric silencing is evolutionarily conserved, and (iii). A deep literature search to identify roles in chromosome segregation that could be generating this critical phenotype. None of the protein-coding candidates we examined passed these filters.

In contrast, the major satellite transcript was the most upregulated transcript in *Dicer1*$^{-/-}$ cells, and activation of major satellite transcription has been shown to cause chromosome segregation defects in mouse cells[56]. The minor satellite transcript on the other hand was very lowly expressed and relatively stable in all conditions (Fig. S8C and D). The abundance of the major satellite transcripts increased dramatically over the culture time-course, but was reduced in viable clones and in *Dicer1*$^{-/-}$ d8 cells with *Brd4* or *Elp3* mutations (Fig. 4C). This transcript was also strongly downregulated in transcriptomes from *Brd4*$^{BD2-/-}$ *Dicer1*$^{-/-}$ and *Elp3*$^{HAT+/-}$ *Dicer1*$^{-/-}$ clonal double mutants (Fig. S8B).). Targeting of the major satellite transcripts with antisense oligonucleotides (ASOs) did not suppress the viability and proliferation defects of *Dicer1*$^{-/-}$ mESCs across a range of concentrations or with strand-specific ASOs tested either individually or in combination (Fig. S7L and M). However, the mutation of *Brd4* strikingly reversed strand-specific transcription of the satellite transcripts in cultured *Dicer1*$^{-/-}$ cells, closely resembling transcription in viable clones that had undergone the strong selection (Fig. 4B; in this figure the data being plotted is a $\log_2$ transformation of the ratio of the abundance of the forward strand to the reverse strand). Along with reduced BRD4 occupancy at satellite loci (Fig. S9A), and a reduction in elongating Pol II (Fig. S9B), chromosomal defects of *Dicer1*$^{-/-}$ cells were also rescued by *Brd4*$^{BD2-/-}$ and *Elp3*$^{HAT+/-}$ (Fig. 4D). We further detected a substantial reduction of RAD21 at both the major and minor satellite loci in *Brd4*$^{BD2-/-}$ *Dicer1*$^{-/-}$ double mutants (Fig. S9C) consistent with recent findings that BRD4 interacts directly with RAD21 in human cells[57] and in *D. melanogaster*[58] as well as with NIPBL in humans[57,59]. *RAD21* encodes cohesin, while NIPBL encodes a component of the cohesin loading complex, which is responsible for proper chromosome cohesion and subsequent segregation at mitosis. Mutants in these and other cohesion complex genes, as well as in Brd4-BD2, underlie Cornelia de Lange syndrome[57,59].

**Dicer-dependent genome stability via BRD4 is deeply conserved**. In quiescent Dicer mutant *S. pombe* cells, viability is

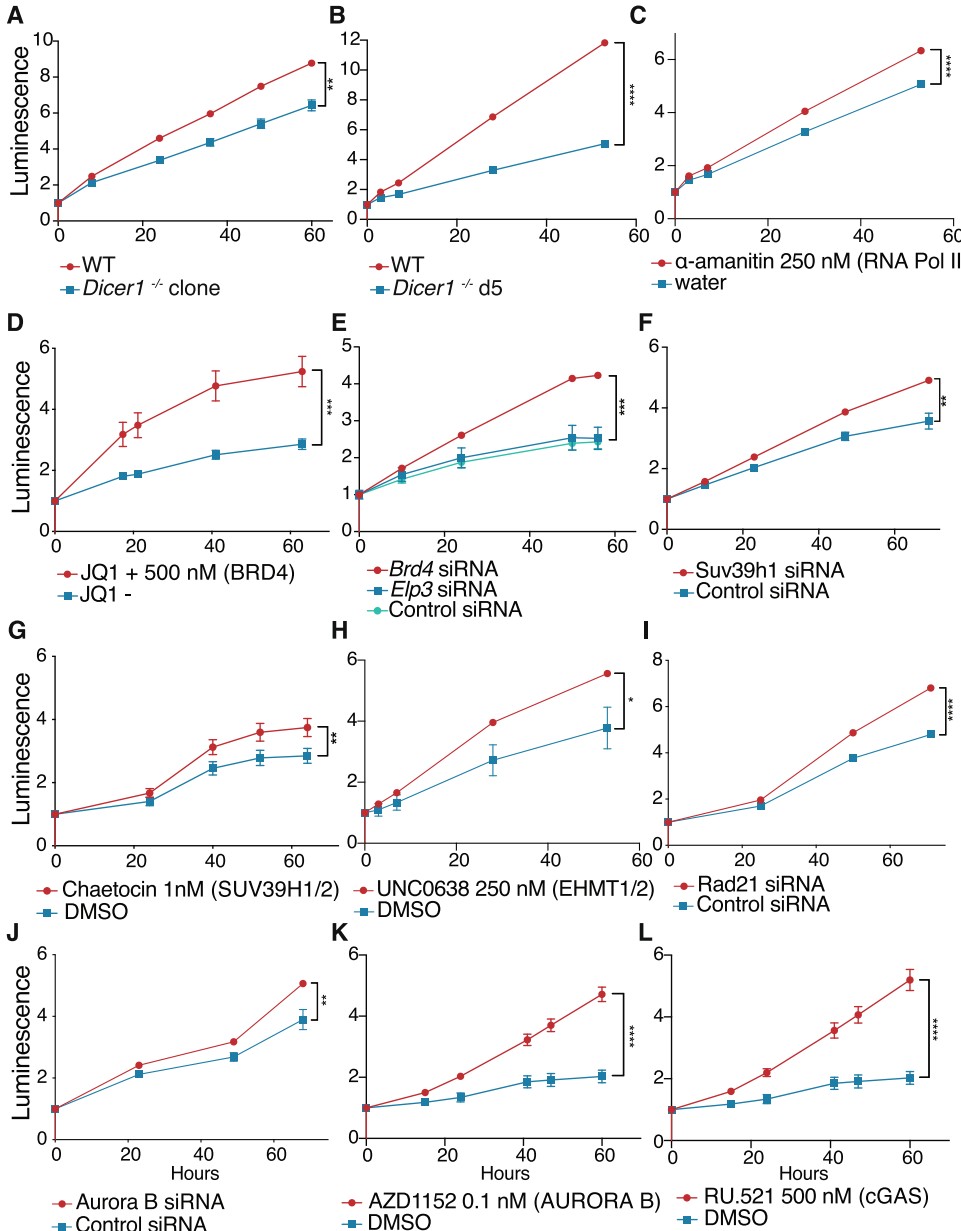

**Fig. 3 Rescue of *Dicer1*$^{-/-}$ mESCs with siRNA and with small molecules targeting suppressors. A**, **B** MT proliferation and viability assay of uninduced wild type, compared with either *Dicer1*$^{-/-}$ clones **A** or *Dicer1*$^{-/-}$ induced cells over days 4–7 **B**. *Dicer1*$^{-/-}$ mESCs have a strong viability defect. **C**–**L** MT assays reveal that the proliferation and viability defect of *Dicer1*$^{-/-}$ induced cells over days 4–7 was rescued by targeting RNA Pol II with α-amanitin **C**, BRD4 with JQ1 **D**, *Brd4* or *Elp3* with siRNAs **E**, *Suv39h1* with siRNAs **F**, SUV39h1/2 with the inhibitor chaetocin **G**, EHMT1/2 with the inhibitor UNC0638 **H**, *Rad21* with siRNAs **I**, *Aurora B* with siRNAs **J**, AURORA B with the inhibitor AZD1152 **K**, or cGAS with the inhibitor RU.521 **L**. Luminescence is plotted over time (hours) after addition of pre-luminescent metabolite at t0. DMSO was used for small molecule delivery and slightly inhibited growth of untreated controls. One representative experiment is plotted in each panel (N = 3 independent biological experiments), errors represent SEM, but may be smaller than points. **** — p-value < 0.0001, *** — p-value between 0.0001 and 0.001, ** — p-value between 0.001 and 0.01, * — p-value between 0.01 and 0.05, two-sided t-test of the final timepoints of replicate experiments. Source Data are provided in the Source Data file.

partially restored by mutation of the H3K9 methyltransferase Clr4 and its interacting partners, as well as the HP1 homolog Swi6, and by viable alleles of the spindle assembly checkpoint Ndc80, required on G$_0$ entry[18]. In mESCs, we found that mutations in the H3K9me3 methyltransferases *Suv39h1/2*, *Ehmt1(GLP)*, and *Ehmt2(G9a)*, were also weak suppressors of *Dicer1*$^{-/-}$ viability defects (Fig. 2A–C). Targeting *Suv39h1* with siRNA (Fig. 3F) and sgRNA (Fig. S7K) significantly alleviated the *Dicer1*$^{-/-}$ viability defect, as did the *Suv39h1* inhibitor chaetocin[60] (Fig. 3G) and the *Ehmt1/2* inhibitor UNC0638[61] (Fig. 3H). The majority of H3K9me2/3 is found at the

pericentromere, and is thought to recruit the chromosome passenger complex, comprising AURORA B kinase, SHUGOSHIN 1, SMC1, SMC3, RAD21, and NIPBL[33]. AURORA B kinase phosphorylates members of the NDC80 complex[62] and centromeric cohesin[63,64]. Both inhibition[65] and overexpression of AURORA B[66] leads to chromosome segregation defects. RAD21 is a core component of the cohesin complex that confers sister chromatid attachment during mitosis and is maintained specifically at the centromere until anaphase[67]. Targeting either of these genes with siRNAs ameliorated the proliferation defects of *Dicer1*$^{-/-}$ cells (Fig. 3I and J), as did targeting AURORA B with the

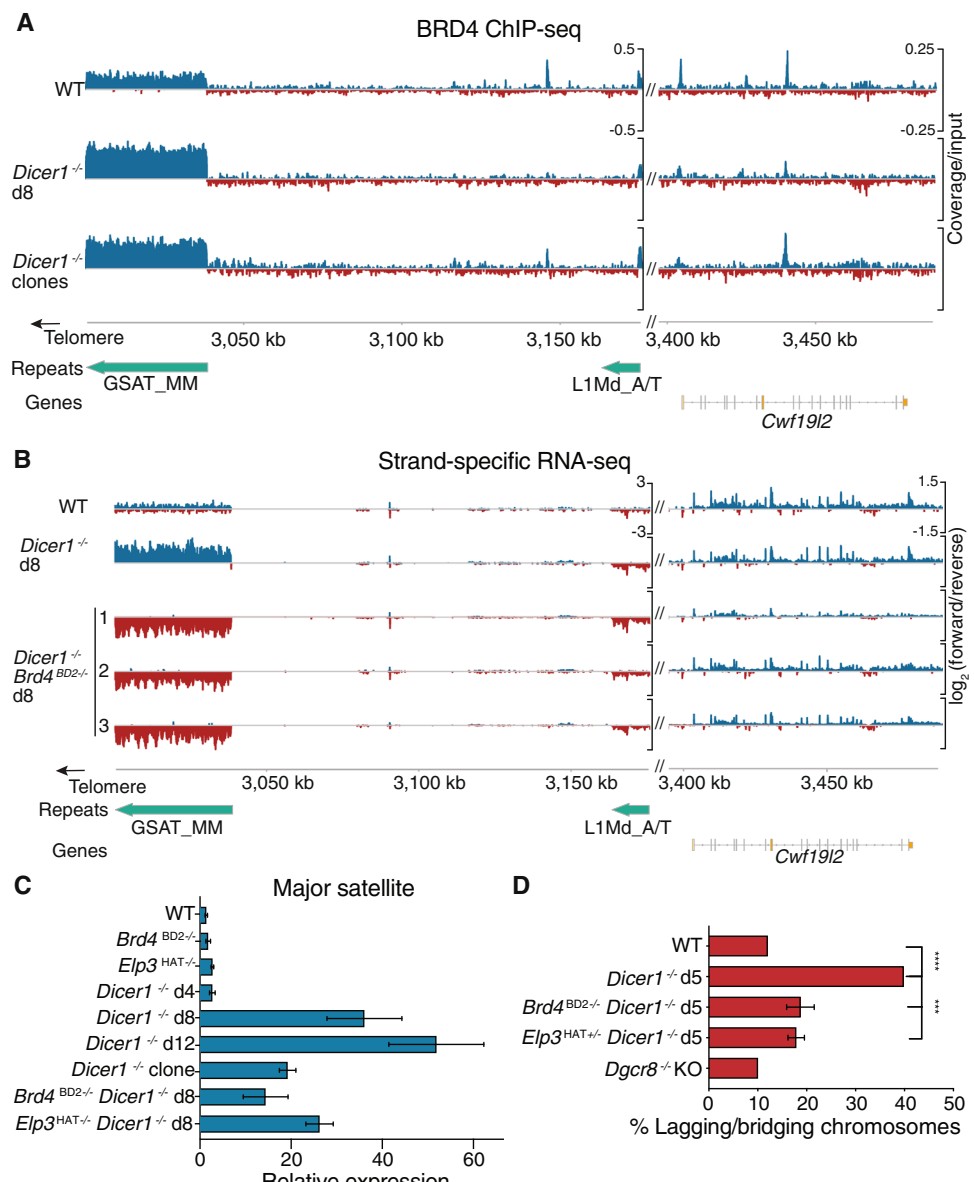

**Fig. 4 BRD4 drives expression of satellite sequences in *Dicer1*$^{-/-}$ mESCs and suppressors reduce chromosome defects. A** BRD4 ChIP-seq coverage normalized to input plotted over the same major satellite repeat region at the end of chromosome 9 as in Fig. 1. BRD4 is enriched over the satellite repeat 8 days after deletion of *Dicer1* (*Dicer1*$^{-/-}$ d8) and in viable *Dicer1*$^{-/-}$ clones. **B** Strand-specific transcriptional upregulation of the major satellite repeat (GSAT_MM) on chr9 is reversed in *Dicer1*$^{-/-}$ *Brd4*$^{BD2-/-}$ cells. The log$_2$ normalized ratio of forward (blue) and reverse (red) strand is plotted. One representative replicate is plotted for WT and *Dicer1*$^{-/-}$ cells 8 days after *Dicer1* deletion (d8). 3 independent *Dicer1*$^{-/-}$ cultures have reversed transcription at the major satellite after *Brd4* mutation in the BD2 domain, 8 days after *Dicer1* deletion. The WT and *Dicer1*$^{-/-}$ d8 cultures are the same replicates from Fig. 1. **C** RT-qPCR confirms a strong accumulation in *Dicer1*$^{-/-}$ mESCs of transcripts from the major satellite, which is reduced in *Brd4*$^{BD2-/-}$ or *Elp3*$^{HAT+/-}$ doubles or *Dicer1*$^{-/-}$ clones. Fold change was calculated relative to *Actb* and uninduced wild type controls. $N = 3$ independent biological experiments ($N = 2$ for Elp3), error bars represent SEM. Source Data are provided in the Source Data file. **D** Chromosomal defects are reduced in *Dicer1*$^{-/-}$ mESCs with *Brd4*$^{BD2-/-}$ or *Elp3*$^{HAT+/-}$ mutations. The WT, *Dicer1*$^{-/-}$ d5, and *Dgcr8*$^{-/-}$ data is reproduced for comparison with Fig. 1B ($n = 100$, **** — $p$-value < 0.0001, *** — $p$-value between 0.0001 and 0.001, Fisher's exact test, two-tailed).

pharmacological inhibitor AZD1152[68] (Fig. 3K), with little effect in wild type cells (Fig. S7). We found that all three classes of suppressors—transcription, H3K9 methylation, and chromosome segregation—significantly reduced the incidence of lagging and bridging chromosomes at anaphase in *Dicer1*$^{-/-}$ mutant clones (Fig. 4D) or in siRNA knockdowns in *Dicer1*$^{-/-}$ cultured cells (Fig. S9D). In fact, we found that inhibiting the cytoplasmic dsDNA sensor cGAS with the small molecule RU.521[69] essentially rescued the viability and proliferation defects of *Dicer1*$^{-/-}$ mESCs (Fig. 3L) at concentrations that reduced viability of wild

type mESCs (Fig. S7J). Thus, the reduction in cell viability observed in *Dicer1*$^{-/-}$ mESCs is due in large part to the generation of cytosolic dsDNA caused by lagging chromosomes in mitosis.

In *S. pombe*, homologs of *Brd4* are encoded by Bdf1 and Bdf2, though Bdf2 more closely resembles *Brd4*. Bdf1 and Bdf2 are genetically redundant in *S. pombe*[70] and the *bdf1Δbdf2Δ* double-mutant is inviable (Fig. S10A), as are *Brd4* knockout mutants in mammalian cells. Perhaps for this reason, mutants in Bdf1 and Bdf2$^{BRD4}$ have not been recovered in fission yeast genetic screens

for Dicer suppressors[18]. We constructed a collection of Bdf2 mutant strains (Fig. S10B) to inactivate specifically BD1 or BD2. Strikingly, only $bdf2^{BD1\Delta}$ displayed synthetic lethality with $bdf1\Delta$ (Fig. S10A), showing that, as in mammalian cells, the essential function of BET proteins is centered on BD1[71]. We generated double deletion mutants of Dicer with $bdf2\Delta$. The double mutant strains suppressed sensitivity to the microtubule poison thiabendazole (TBZ) (Fig. 5A), the accumulation of pericentromeric transcripts (Fig. 5C), and lagging chromosomes (Fig. 5D). Furthermore, because chromosome segregation defects cause a loss of viability on $G_0$ entry, we found higher viability of the $dcr1\Delta bdf2\Delta$ double mutants (Figs. 5B and S11A). We constructed a yeast mutant equivalent to our $Brd4$ CRISPR alleles in mammalian cells—a small in-frame deletion of 7 amino-acids in BD2 (Fig. 5E)—which was fully viable in the $bdf1\Delta$ background (Fig. S10A and B). Strikingly, this $dcr1\Delta$ $bdf2^{CR-BD2}$ double-mutant was comparable to $dcr1\Delta$ $bdf2\Delta$ in TBZ sensitivity, $G_0$-entry, and pericentromeric silencing (Figs. 5A–C and S10C). The triple-mutant $dcr1\Delta$ $bdf1\Delta$ $bdf2^{CR-BD2}$ displayed nearly complete suppression of $dcr1\Delta$ phenotypes, with loss of TBZ sensitivity, near-wild-type viability on $G_0$-entry, and transcriptional silencing of pericentromeric repeats. As in mammalian cells, pericentromeric transcripts in Dicer mutants displayed strong strand-specificity. However, in the triple mutant we found that the ratio of the forward strand to the reverse strand was much closer to equal (Fig. 5C), indicating strand specificity is reversed in this strain. These results indicate that Dcr1 centromeric function is deeply conserved in mESCs and fission yeast, both in cycling cells and upon $G_0$ entry, and distinguishes between bromodomains BD1 (essential function) and BD2 (silencing function) of the transcription factor BRD4.

## Discussion

In fission yeast, Dicer acts on RNA substrates to release Pol II from the pericentromere[14,15,72] and from other sites of collision between the transcription and replication machineries[12,13]. In ES cells, our data suggest that the viability defect of $Dicer1^{-/-}$ cells is a consequence of transcription of the centromeric satellite repeats, and we have shown that this defect can be rescued by hypomorphic mutations in transcription factors $Brd4$ and $Elp3$ or by inhibiting Pol II. In support of our findings, nuclear-localized DICER1 has been found associated with satellite repeats and their transcripts, and with the nuclear protein WDHD1 in complex with Pol II[29], and regulates transcription at this locus in multiple mouse cell types[29,73]. Dicer has been reported to interact with Pol II in *Drosophila*[74] and in human HEK293 cells, where it also prevents the accumulation of dsRNAs from satellite repeats[75]. Additionally, there are well-described interactions between Dicer and the transcriptional machinery in *S. pombe*[12,76,77].

The fact that $Brd4$ and $Elp3$ were the strongest suppressors of the $Dicer1^{-/-}$ phenotype, which was also suppressed with low doses of $\alpha$-amanitin, strongly implies transcription as the underlying mechanism. We found strand-specific accumulation of major satellite transcripts in $Dicer1^{-/-}$ mESCs that was reversed by $Brd4^{BD2-/-}$ mutations and by selection for viability that generated our clonal lines. Similar reversals have been observed on transition between $G_1$ and $S/G_2$ phases of the cell cycle in mouse embryos up to the 4-cell stage, and so transcript reversal likely reflects the exit from $G_1$ arrest in $Dicer1^{-/-}$ ES cell clones[78]. Transcription of the reverse strand may be toxic because of collision with DNA replication, which may occur predominantly in one direction as the repeats themselves have inefficient origins of replication[12]. Intriguingly, BRD4 interacts with PCNA, which is highly enriched on the lagging strand[79]. This could account for strand specificity of BRD4-dependent

transcripts, which DICER1 would normally remove from the lagging strand to avoid collisions during replication. We considered the possibility that the major satellite transcripts generate DNA:RNA hybrids or R-loops that lead to deleterious effects in $Dicer1^{-/-}$ mESCs. However, our strongest suppressor, $Brd4$, has been shown to prevent the accumulation of R-loops[80,81] and so mutations would likely only further increase the accumulation of R-loops in $Dicer1^{-/-}$ cells. In the absence of RdRP, and consistent with the lack of small RNAs, there was no decrease in H3K9me2/3 at the pericentromere, while there was a decrease of H3K9me3 at retrotransposons that have corresponding small RNAs[44].

We used our suppressors to further dissect the mechanism underlying the $Dicer1^{-/-}$ phenotype. We found a direct relationship between transcription (BRD4) and cohesion (RAD21/AURORA B) that suggests transcriptional suppressors may act through the cohesin complex, as in *S. pombe*[82,83]. Ectopic expression of satellite transcripts in mouse and human cells has been found to cause chromosome mis-segregation and genome instability in tumors[56]. The mechanism is thought to involve interaction of major satellite transcripts with the MCM complex required for replication, and replication stalls in the S phase of the cell cycle[56], very much like the $Dicer1^{-/-}$mESCs analyzed here. These observations suggest a re-evaluation of the high frequency of $Dicer1$ mutants in cancer[84–86], in which chromosomal abnormalities lead to increased genome instability. We find that chromosomal phenotypes activate cytoplasmic DNA sensors, such as cGAS/STING, which respond to the presence of micronuclei[87] and generate a type I interferon response that reduces proliferation and leads to apoptosis[88]. Our data indicates this mechanism is the cause of the proliferation and viability defects of $Dicer1^{-/-}$ mESCs.

Aberrant accumulation of major satellite RNA has also been shown to recruit heterochromatin factors SUV39H1[89–91] and HP1[92] to the pericentromere. However, the mutation of $Brd4$, which reduces transcript levels significantly, did not result in loss of H3K9me3. Instead, H3K9me3 accumulation in $Dicer1^{-/-}$ mESCs more closely resembles the phenotype of yeast $dcr1\Delta$ quiescent cells in which rDNA accumulates H3K9me2 and relevant small RNAs are not detected[18]. The arrest of $Dicer1^{-/-}$ ES cells in $G_1$ suggests that they may enter a non-dividing $G_0$-like state that affects viability. Major satellite transcripts accumulate primarily in late $G_1$[78,93] and $G_1$ arrest is a prerequisite for entry into $G_0$ in fission yeast and mammalian cells[86,94,95]. If the loss of DICER1 does force the cells into a suspended $G_1$ or $G_0$-like state and DICER1 is required to maintain this state as it is in *S. pombe*, then second site mutations or epimutations would be required to exit this state and resume growth[80,81].

We have identified suppressors of the $Dicer1^{-/-}$ ES cell proliferation, viability and chromosomal defects in transcriptional activators, H3K9 methyltransferases, and chromosome segregation factors. These same classes of suppressor mutations rescued viability defects in Dicer deletion mutants in *S. pombe*[18]. The most significant upregulated non-coding RNA in $Dicer1^{-/-}$ mESCs was the major satellite RNA, directly comparable to pericentromeric transcripts in *S. pombe*. Thus, the mechanism by which Dicer regulates transcription, even in the absence of RdRP and small RNAs, may be conserved from fission yeast to mammals. The different role played by small RNA and Argonaute proteins in these two model organisms may reflect distinct mechanistic details underlying this function, with Dicer as a key factor in the process. The role we identified for BRD4 in pericentromeric transcription was also conserved in *S. pombe*. Strikingly, this conservation held true for specific bromodomains, with Bdf2-BD1 being essential in the absence of Bdf1, while Bdf2-BD2 is involved in suppression of $dcr1\Delta$. Mutants in the

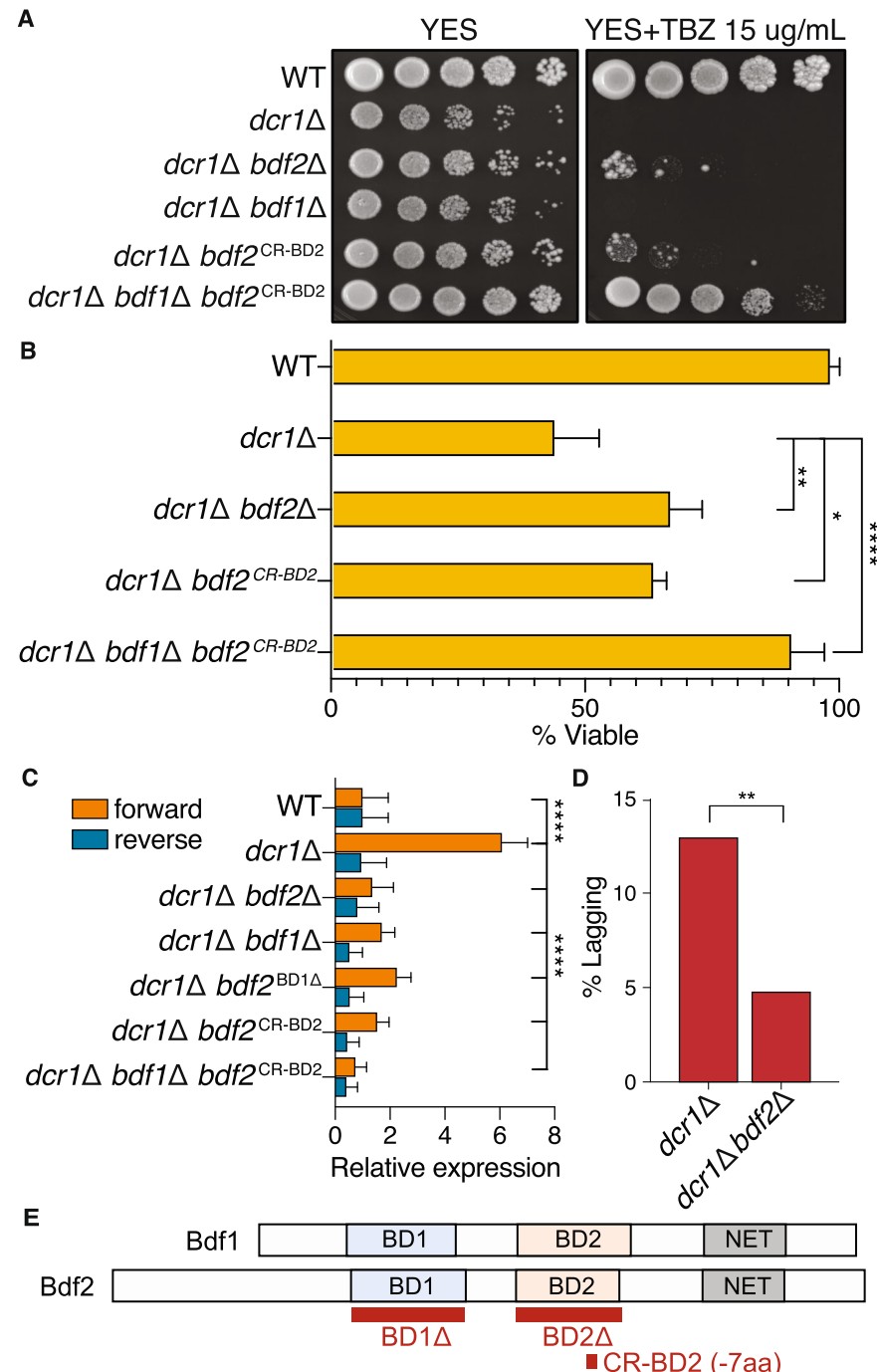

**Fig. 5 The genetic interaction of *Dicer1* with *Brd4* is conserved in *S. pombe*. A** Spot growth assays (10-fold dilutions) on media supplemented with the microtubule poison thiabendazole (TBZ). Mitotic defects of *dcr1Δ* are partially suppressed by *bdf2Δ* and *bdf2*^CR-BD2, restoring viability, and fully suppressed in the triple-mutant *dcr1Δ bdf1Δ bdf2*^CRBD2, which mimics the 7 amino acid deletion in BD2 found in *Dicer1*^−/− ES cells (Fig. 2E). **B** The loss of viability at G₀-entry seen in dcr1Δ mutants is also suppressed in *dcr1Δ bdf2Δ* and *dcr1Δ bdf2*^CR-BD2, and is fully rescued in the triple-mutant *dcr1Δ bdf1Δ bdf2*^CRBD2. Barplots represent the mean, error bars represent SD. **** − p-value < 0.0001, ** − p-value between 0.001 and 0.01, * − p-value between 0.01 and 0.05, two-sided t-test. (N = 4 independent biological experiments). Source Data are provided in the Source Data file. **C** Forward strand *dh* pericentromeric transcripts accumulate to high levels in *dcr1Δ* mutants relative to the reverse strand, but are strongly reduced in double mutants with *bdf1Δ* and *bdf2Δ*, reversing strand preference. Silencing was fully restored in *dcr1Δ bdf1Δ bdf2*^CRBD2. **** − p-value < 0.0001 – two-sided t-test. (N = 3 independent biological replicates). **D** Knockout of Bdf2 in *bdf2Δ* reduces the lagging chromosome phenotype of *dcr1Δ* mutants. ** − p-value between 0.001 and 0.01 – Fisher's exact test, two-tailed, N = 100 cells. **E** Schematic representation of BD1 and BD2 deletions used.

centromeric H3K14 acetyltransferase Mst2 suppress $dcr1\Delta$ in a similar fashion[18,96], and H3K14A mutants lose pericentromeric silencing[97]. In *S. pombe*, $bdf2\Delta$ mutants alleviate DNA damage accumulation at the S-phase checkpoint, and suppress hydroxyurea sensitivity in checkpoint mutants[70]. $dcr1\Delta$ mutants accumulate stalled RNA pol II and DNA damage at S phase[12], and suppression by $bdf2\Delta$ results in reduction of reverse-strand pericentromeric transcription, lower amounts of stalled RNA pol II and reduced DNA damage. This allows maintenance of heterochromatin through the cell cycle and ensures its mitotic inheritance, as in $mst2\Delta$[96]. In cancer cells, BD1 is also essential, while BD2 has specific roles in inflammation, autoimmune disease, and specific cancer subtypes. Moreover, BD2 specific inhibitors are promising therapeutic agents for these conditions with minimal side effects[98]. BD1 has higher affinity for acetylated lysines on histone H4, while BD2 has a higher affinity for H3 acetyl lysines[99], consistent with a key role for H3 modifications in centromeric silencing in *S. pombe*. These conserved genetic and mechanistic interactions with transcription, DNA replication, histone modification, and sister chromatid cohesion likely contribute to "DICER1 syndrome", in which Dicer mutations predispose cancer and viral infection[86]. A potential therapeutic role for BRD4 and BD2 inhibitors is suggested by these interactions.

## Methods

**Cell lines and tissue culture**. Mouse embryonic stem cells, including *Dicer1*$^{flx/flx}$ lines, were grown in 2i conditions. The 2i medium consists of: 250 mL Neurobasal medium (Gibco 21103-049), 250 mL DMEM/F12 (Gibco 11320-033), 2.5 mL N2 supplement (Gibco 17502-048), 5 mL B27 supplement (Gibco 17504-044), 5 mL GlutaMAX (Gibco 35050-061) and 3.3 mL 7.5% BSA Fraction V (Gibco 15260-037). Just prior to adding the media to the cells, the following additional supplements were added: PD0325901 to 1 $\mu$M (Stemgent 04-0006), CHIR99021 to 3 $\mu$M (Stemgent 04-0004), 2-Mercaptoethanol to 100 $\mu$M (Gibco 21985-023), and Mouse LIF to $10^3$ units/mL (EMD Millipore ESG1106). Culture dishes were coated with gelatin (1X Attachment Factor – ThermoFisher S006100) for a minimum 15 min, gelatin was removed, and dishes were allowed to dry before cells were plated. Cells were split with TrypLE Express Enzyme (Gibco 12605-010) for 5 min at 37 °C. Cells were collected in enough media for a 10-fold dilution of the TrypLE Express and pipetted up and down a minimum of 10 times to achieve a single cell suspension. Cells were then spun for 5 min at 150 × g and resuspended in media for counting and plating. Cells were frozen in freezing media: 10% DMSO (Sigma D2650), 50% ES-FBS (Gibco 16141-002), and 40% cells in media. Inducible *Dicer1* knockout cells were treated with hydroxytamoxifen (OHT – Sigma H7904) at a final concentration of 1 $\mu$M. For immunofluorescence, cells were grown on poly-L-lysine coated coverslips in 6 well dishes (Corning 354085).

Lenti-X 293 T cells were grown in DMEM with 10% FBS and pen/strep. All cell lines used in this study were grown at 37 °C and 5% $CO_2$. All mouse embryonic stem cells used are male.

**Transfections**. In the case of siRNA transfection, the Lipofectamine RNAiMAX was used (ThermoFisher 13778030) according to the manufacturer's protocol. The siRNAs were the OnTarget Pools targeting genes of interest from Dharmacon. In the case of Cas9/sgRNA transfection, Lipofectamine CRISPRMAX (ThermoFisher CMAX00001), TrueCut Cas9 (ThermoFisher A36497), and TrueGuide sgRNAs (ThermoFisher custom) were used. The Lipofectamine CRISPRMAX protocol was followed for transfection preparation. In all cases, transfection reagents were prepared ahead of time and added to wells. Cells were then split and added to the transfection-containing media (reverse transfection).

**DNA/RNA isolation and cDNA preparation**. Cells were collected by splitting as described and pelleted. To isolate genomic DNA, the Purelink Genomic DNA Mini Kit (K182001) was used as directed. Genomic DNA was eluted in TE and stored at −20 °C. Concentration was measured with the Nanodrop. RNA isolation was performed on either fresh pellets or fresh pellets were resuspended in the appropriate amount of TRIzol (ThermoFisher 15596018), snap frozen, and stored at −80 °C until the protocol could be finished. Cells were resuspended in the appropriate amount of TRIzol and the manufacturer's protocol was followed as directed, save for the use of 80% ethanol for the washing steps to increase retention of small RNAs. RNA was resuspended in ultra-pure water and stored at −80 °C. RNA concentration was measured by Nanodrop and quality was measured with the Bioanalyzer. For cDNA preparation 1 microgram aliquot of pure RNA was DNase treated and reverse transcribed using the Superscript IV Vilo ezDNAse kit (ThermoFisher 11766050).

**RT-qPCR and ChIP-qPCR**. In general qPCR was performed with Taqman probes. For qPCR on genomic DNA, genomic DNA was quantified using Nanodrop or Qubit and diluted to 5 nanograms per microliter. The Taqman CNV Assay and *Tfrc* CNV Assay control were run together in triplicate in 10 microliter reactions on a 384 well plate with the 2X Genotyping Mastermix (ThermoFisher 4371355) and analyzed with the △△Ct method. For qPCR on cDNA, the cDNA was generally diluted at least 1 to 20 and 1 microliter of diluted cDNA was used in 10 microliter reactions on a 384 well plate in a reaction with the Taqman Assay, *Actin* or *Gapdh* control assay with a different fluorophore, and the Taqman Fast Advanced Master Mix (ThermoFisher 4444557). The run was analyzed with the △△Ct method. For ChIP-qPCR, ChIP DNA was diluted along with the H3 and Input controls at least 1 to 10. For each run a standard curve was run to test efficiency of the primers. The reactions were 10 microliters in 384 well plates with PowerUp SYBR Green Master Mix (ThermoFisher A25918). Each run was tested for efficiency and then a percent input calculation was used to determine enrichment.

**Western Blot**. Cell pellets were washed twice with PBS and resuspended in an appropriate amount of RIPA buffer (ThermoFisher 89900) and incubated for 15 min on ice. Samples were centrifuged at $14,000 \times g$ for 15 min and the supernatant was transferred to a new tube. Protein concentration was measured with a microplate BCA assay in triplicate (Pierce 23252). Equal amounts of protein (at least 15 micrograms) was loaded on a 4–20% gradient gel (Bio-Rad 4561096) and run until loading dye reached the bottom of the gel. The proteins were transferred onto nitrocellulose membranes using the Trans-Blot Turbo Transfer system. A Ponceau stain was then performed to verify the transfer. Gels were often divided to probe for different sized proteins. Blocking was done in 5% Milk in TBST. Primary antibodies were incubated, at a dilution of 1:1000, for at least an hour at room temperature or overnight at 4 °C. Washes were performed with TBST. The secondary antibody, HRP goat anti-rabbit (Abcam ab97051), was incubated for at least 1 h at room temperature. The secondary antibody was detected with the SuperSignal West Pico Plus Chemiluminescent Substrate (ThermoFisher 34577) and imaged with the Bio-Rad ChemiDoc MP. In the need for stripping and re-probing, Restore Western Blot Stripping Buffer was used (ThermoFisher 21059). The primary antibodies used were beta-tubulin (Cell Signaling 2146) and γH2A.X (Cell Signaling 2577) for Fig. S1, and BRD4 (Active Motif 39910), ELP3 (Cell Signaling 5728) and actin (Cell Signaling 3700) for Fig. S6.

**Cell proliferation and viability assays**. For cell proliferation and viability assays the Promega RealTime-Glo MT Cell Viability Assay (Promega G9711) was used. The assay was used according to the protocol, with the two reagents diluted to 1X in a white 96 well plate that had been coated with 1X Attachment Factor. Cells were then added to well and allowed to stabilize for at least 1 h before the first reading. Luciferase readings were performed periodically over a period of 72 h in a 96 well plate reader at 37 °C with a 250 millisecond integration time at a height of 1 millimeter above the plate. Readings were done in at least 5 sites in the well to minimize noise. The assay was analyzed by normalizing all readings to the initial reading (relative luminescence in the figures) and each timepoint was plotted over time.

**BrdU cell cycle analysis**. BrdU Flow Cytometry Cell Cycle Analysis was performed with a FITC BrdU Flow Kit (BD Pharmingen 559619). Cells were cultured normally and then dosed with an appropriate amount of BrdU for at least 1 h prior to harvest. The Flow Kit protocol was followed as directed and the Bio-Rad SE3 Cell Sorter was used.

**Immunofluorescence staining and quantification**. Cells were grown on glass coverslips in 6 well dishes. Media was removed, cells were washed twice with PBS, and then treated with 4% paraformaldehyde in PBS for 10 min at room temperature. Cells were washed 3 times in PBS and treated with a quenching solution (75 mM $NH_4Cl$ and 20 mM Glycine in PBS) for 10 min. Fixed coverslips then permeabilized with PBS-Triton X-100 0.1% for 2 min on ice. Blocking and antibody dilutions were performed with 5% BSA. The primary antibody was diluted at a ratio of 1:500 and applied for 1 h at room temperature. The secondary antibody was diluted as direction and applied for 45 min at room temperature. Prolong Gold with DAPI (ThermoFisher P36931) was used for mounting. The primary antibodies used were anti-CENPA (Cell Signaling 2047), anti-tubulin (Cell Signaling 2146) and anti-H3K9me3 (Abcam ab176916), and the secondary antibodies used were donkey anti-rabbit Alexa Fluor 488 (Thermo A-21206) and goat anti-rabbit Alexa Fluor 594 (Thermo A-11012).

**Cloning sgRNA and Cas9 plasmids**. In order to generate a lentiviral Cas9 construct, we cloned a PGK promoter and mCherry fluorescent reporter into the Lenti-Cas9-puro vector (gift from Ken Chang/Vakoc Lab). In order to generate sgRNA lentiviral vectors we first ordered sense oligos with a CACCG 5′ overhang and antisense oligos with a AAAC 5′ overhang and a C 3′ overhang. The oligos were phosphorylated and annealed with T4 PNK (NEB M0201S) in a thermocycler with the following conditions: 37 °C for 30 min and then a ramping down from 95 °C to 25 °C at 5 °C per minute. Annealed oligos were cloned into a Bsmb1

digested Lenti-sgRNA-GFP-LRG plasmid (gift from Ken Chang/Vakoc Lab) with T4 DNA ligase (NEB M0202S) for 30 min at room temperature. A 1:200 dilution of annealed oligo was used (1 uL) with 25 nanograms of digested plasmid (1 uL) in a 10 microliter ligation reaction. Lentiviral plasmids were transformed into Stbl3 (ThermoFisher C737303).

**Lentiviral production and infection**. The PSPAX2, VSVG, and transfer plasmids were prepared according to standard bacterial culture practices in Stbl3 (ThermoFisher C737303). General bacterial culture practices were followed except the cultures were grown at 30 °C overnight to reduce recombination. Lentiviruses were made with Lenti-X 293 T cells (Takara 632180) cultured in DMEM with 10% FBS and pen/strep. Prior to transfection, cells were plated on a 15 centimeter dish coated with gelatin as described. Cells were grown until 90% confluence. Transfection of PSPAX2, VSVG, and transfer plasmids was performed with Lipofectamine 3000 according to the forward transfection protocol. Lentiviral containing media was collected at two timepoints depending on the yellowing of the media. The lentivirus was then isolated from the spent media using the Lenti-X Concentrator (631231) and following the protocol as directed. Pelleted virus was resuspended in 2i media, aliquoted, snap-frozen on dry ice, and stored at −80 °C. Infections in ES cells were performed by adding the concentrated lentivirus to attached cells with 8 ug/mL of DEAE-Dextran (Sigma D9885). Lentivirus was removed after 24 h and then the cells were cultured normally.

**CRISPR screen**. The Chromatin Modifier sgRNA library was prepared by spreading and growing the bacteria on large agar plates. The plates were scraped and a Qiagen Maxiprep (Qiagen 12162) was performed to isolate the lentiviral plasmids. Lenti-X 293 T cells were transfected with the library as described and lentivirus was collected with the Lenti-X concentrator as described. Six replicate 15 centimeter (cm) dishes of ES cells were then infected with the concentrated lentiviral particles at a calculated representation of 500 events for each sgRNA. After 2 days the six dishes were split into two separate 15 cm dishes each and one of the two was treated with OHT to induce the *Dicer1* mutation. At this time, the GFP% in the population was monitored to ensure the infection was at a multiplicity that would produce on average less than 1 sgRNA per cell. The individual 15 cm dishes were grown over a period of 3 weeks. Each time the dishes were split genomic DNA was collected for sgRNA amplification. At the end of the growth period genomic DNA was isolated with the Qiagen Blood and Tissue Kit (Qiagen 69504). The sgRNA loci were amplified from the amount of genomic DNA necessary to maintain the representation of the library ~500x using the high fidelity Q5 polymerase (NEB M0492) and Illumina-compatible barcoded primers were used in a second PCR to create libraries for each replicate at each timepoint. Libraries were quantified with the KAPA Illumina Quantification Kit (KAPA KK4824).

**Single sgRNA infection and flow cytometry**. Single sgRNA vectors were constructed and lentiviral particles were generated as described. The Cas9-expressing *Dicer1* flx/flx cell line was infected as described before. Two days after infection, the cells were split. At this time one third of the cells were plated and kept cultured as normal, one third of the cells were induced to mutate *Dicer1* with OHT for 24 h, and one third of the cells were washed in PBS, run over a strainer cap flow cytometry tube, and analyzed for EGFP signal on the Bio-Rad SE3 flow cytometer. The cells were then cultured as one *Dicer1* wild type population and one *Dicer1* mutant population and every four days the cells were split and cells were again analyzed by flow cytometry for EGFP signal. At the end of the timecourse genomic DNA was collected as described for amplicon sequencing.

**Amplicon sequencing library preparation**. Genomic DNA from single sgRNA lentiviral timecourse experiments was isolated as described. Primers flanking the sgRNA-targeted region that contained Illumina sequence overhangs were used in a PCR reaction with genomic DNA and Q5 high fidelity polymerase (NEB M0492). A second PCR with primers containing barcodes and targeting the Illumina overhangs of the first PCR was performed with a minimal number of cycles to avoid amplification bias. The PCR-generated libraries were gel purified to remove primers. Libraries were quantified with the KAPA Illumina Quantification Kit (KAPA KK4824).

**Small RNA sequencing library prep**. Small RNAs were isolated from total RNA using Novex TBE-Urea Gels (ThermoFisher EC6885BOX). Libraries were constructed with the TruSeq Small RNA Kit (Illumina RS-200-0012). Libraries were quantified with the KAPA Illumina Quantification Kit (KAPA KK4824).

**Whole genome sequencing library preparation**. Genomic DNA was isolated as before. DNA was fragmented with Covaris ultrasonication with default settings for the generation of fragments of 350 basepair average size. Libraries were constructed with the TruSeq DNA PCR-Free Kit (Illumina 20015962). Libraries were quantified with the KAPA Illumina Quantification Kit (KAPA KK4824).

**RNA sequencing library preparation**. RNA was isolated with TRIzol as described and DNase treated. When total RNA was sequenced, 1 microgram RNA was first depleted for ribosomal RNA using the rRNA Depletion Kit (NEB E6310). Total RNA libraries were then prepared using the NEBNext Ultra II Directional Kit (NEB E7760). Libraries were quantified with the KAPA Illumina Quantification Kit (KAPA KK4824).

**ChIP and ChIP sequencing library prep**. Cells were grown in 2i conditions as detailed on 15 centimeter plates. A total of 37% formaldehyde is added to a final concentration of 1% in the media and incubated at room temperature for 10 min. The reaction was quenched with for 5 min at room temperature with the 10× glycine buffer from the SimpleChIP Kit (Cell Signaling 56383). The protocol for the SimpleChIP Kit was followed as directed. Chromatin was fractionated using the 1 mL millitubes on the Covaris with optimized custom settings to achieve an average size of 250 basepair fragments. ChIP was performed with recommended dilutions of antibodies and control H3 antibodies. An input control was also generated for each sample. ChIP-sequencing libraries were constructed with the NEBNext Ultra II DNA Library Prep Kit (NEB E7645S). Libraries were quantified with the KAPA Illumina Quantification Kit (KAPA KK4824) and checked for quality on the Agilent Bioanalyzer.

**S. pombe strains and crosses**. *S. pombe* strains were cultured in standard conditions at a temperature of 30 °C. De novo deletion mutant strains were generated by multiplex PCR in which the first PCR (*bdf1Δ*: L1-L2 and L5-L6) run generates ~200 nucleotides of homology regions and a second PCR step generated the final construct (L1-L6 PCR in the presence of the *NatMX* cassette). For *bdf2* point mutants and partial deletion mutants, the equivalent PCR fragment was generated by a multiplex PCR of the L0-M1R and M1F-L10 PCR products (for point mutants) or of the L0-D1R and D1F-L10 PCR products (for partial deletions), followed by multiplex PCR (L1-L12 PCR in the presence of the *HphMX* cassette). All PCRs for strain construction were performed using Phusion High-Fidelity DNA polymerase (New England BioLabs M0530), and oligonucleotides used for strain construction are provided in Supplementary Data 5. PCR products were cleaned by Qiagen PCR Purification (Qiagen 28104) and 1 microgram of DNA was used per transformation using the Frozen-EZ kit (Zymo T2001). All constructed *bdf2* mutant strains were verified to harbor the correct mutations (or partial deletions) by Sanger sequencing. A list of *S. pombe* strains used in this study is provided in Supplementary Data 4. Crosses were performed on malt extract media (ME) at room temperature, followed by random spore analysis, except for the investigation of synthetic lethality between *bdf1Δ* and *bdf2* mutant strains where tetrad analysis was performed. In each tetrad analysis experiment, a minimum of 50 tetrads were isolated and dissected on YES plates, using a Singer MSM400 micro-manipulator (Singer Instruments). Viable colonies were genotyped, and only tetrads where the genotype of every spore could be observed or inferred were taken into consideration for measuring the viability of each genotype; this assay showed complete synthetic lethality (no recovered viable double-mutant) of the *bdf1Δ bdf2Δ* and *bdf1Δ bdf2*BD1Δ genotypes, and no viability defect in the other double-mutants.

**S. pombe assays**. In order to induce G₀, cells were cultured in EMM (Edinburgh minimal medium) and then shifted to EMM-N (Edinburgh minimal medium without nitrogen). Viability was determined 24 h after G₀ entry by isolating ≥100 single-cells on a YES plate, using a Singer MSM400 (Singer Instruments), and measuring their ability to reform a colony, as previously[18]. The presence of mis-segregation in G₀ entry was determined by determining the proportion of rod-shaped cells in the culture (24 h G₀) with a hemocytometer, counting ≥100 cells. Thiabendazole (TBZ) assays were performed on YES plates containing 15 μg/ml of TBZ (Sigma). Cells were grown to log phase, counted, and spotted in equal numbers across a range of dilutions. RT-qPCR for the *dg/dh* repeats was performed by first isolating RNA from log phase cells with the Quick-RNA Fungal/Bacterial Kit (R2014). RNA was then DNase treated and reverse transcribed with the Superscript IV Vilo ezDNase Kit (ThermoFisher 11766050), using centromere-specific primers (p30-F and p30-R) for the strand-specific RT-qPCR, and random hexamers for non-specific RT-qPCR. The resulting cDNA was then used for qPCR using iQ SYBR Green mix (Bio-Rad), using p30 and *act1* primers in the non-stranded reaction, and p30 in the stranded reaction.

**Image quantification**. Intensity of immunofluorescence staining was quantified by first acquiring at least 5 Z-stacks with identical acquisition conditions for each sample across multiple coverslips. Fiji image analysis software[100] was used to calculate a maximum intensity projection and then measure the mean fluorescence intensity across all nuclei in an image. Lagging and bridging phenotypes were quantified by assaying each anaphase for either phenotype by visual inspection.

**Cell proliferation and viability assays statistical analysis**. All statistical analyses comparing treatment were performed using GraphPad Prism software (version 8 and 9). Unpaired two-tailed *t*-tests were used to compare data consisting of two population means at the final timepoints of replicate assays.

**CRISPR screen sequencing analysis**. Sequences were de-multiplexed and the adapters were trimmed from both ends with Flexbar[101] (version 3.5.0). Mapping was performed with Bowtie[102] (version 1.3.0) on a Bowtie index built with the sgRNA library sequences with the default settings and '–norc' to prevent alignment to the reverse complement of the sgRNA sequences. MAGeCK analysis package (version 0.4) was used to determine sgRNA enrichment/depletion as well as gene ranking in the screen[103].

**Amplicon sequencing analysis**. Reads were trimmed with Trimmomatic[104] (version 0.4) to remove all excess Illumina adapter. Reads were collapsed using the FastX toolkit (version 0.0.14) and analyzed for CRISPR-generated mutations with a custom script.

**Small RNA sequencing analysis**. Raw reads were first trimmed with Cutadapt[105] (version 3.4) to clip Illumina 3′ adapters, while retaining the 4nt-long degenerate sequences at the 5′ and 3′ end of each insert. Reads with an insert size between 12 and 60 nt were subsequently filtered using Gordon Assaf's FASTX-Toolkit (https://hannonlab.cshl.edu/fastx_toolkit/) to discard low-quality reads with Phred scores less than 20 in 10% or more nucleotides. PCR duplicates were removed using PRINSEQ[106] (derep option 1, version 0.20.4) and degenerate sequences were extracted and appended to the name of each read using UMI-tools[107] (version 1.0.0). Preprocessed reads were first aligned to calibrator sequences and then either to *Mus musculus* mm10 UCSC genome using Bowtie2[108] (version 2.4.3), which assigns multi-mapping reads in an unbiased way. Aligned reads were filtered for 0-2 mismatches using Samtools[109] (version 1.12) and BamTools[110] (version 2.5.0). For ERV and GSAT (major satellite) analysis, aligned reads were intersected with RepeatMasker annotation (http://www.repeatmasker.org) using BEDtools[111] (version 2.29.2). Read counts were normalized to total calibrator per library. Shaded regions in each panel display minimum and maximum values of three biological replicates for each genotype. Data visualization was performed using R.

**Whole genome sequencing analysis**. Illumina reads were trimmed for adapters with Trimmomatic[104]. Reads were mapped with Bowtie2 with default settings to the mm10 genome[108]. Samtools[109] was used to convert, sort, and index bam files. Freebayes[112] (version 1.3.4) was used with default settings to call SNPs in sequencing data relative to the reference *Mus musculus* mm10 genome. SNPs were filtered for quality and SnpEff[113] (version 4.3) was used to annotate. Filtered and annotated SNPs were also intersected with PFAM domains with BEDtools[111]. Normalized coverage was calculated using deepTools[114] (version 3.5).

**RNA sequencing analysis**. Reads were trimmed for adapters and quality with Trimmomatic[104]. Mapping to the mm10 genome was performed with the STAR aligner[115] with–winAnchorMultimapNmax and–outFilterMultimapNmax set to 100 in order to allow reads to multimap for downstream analysis (version 2.7.7). The analysis of differential expression of both genes and repetitive elements was performed with TEtranscripts[116] (version 2.2.1) and DESeq2[117] (version 3.11). Gene set enrichment analysis was performed with GSEA[118,119] (v4). Intersections between RNA-seq data sets and ChIP-seq datasets were performed with custom scripts. For strand-specific analysis, trimmed reads were mapped with bowtie2 with default settings and -k 5 to allow for multimapping. Normalized, strand-specific coverage was generated using deepTools[114].

**ChIP sequencing analysis**. Reads were trimmed for adapters and quality with Trimmomatic[104]. Bowtie2[108] was used for mapping with default settings and –dovetail, -X 600, and -k 5. The resulting SAM files were converted, sorted, and indexed with SAMtools[109]. Normalized coverage was calculated for library size for each sample and a relative normalized coverage was calculated for IP over input using deepTools[114]. Peaks were called with default settings with MACS2[120] (version 2.2.5). Differential peaks were called with default settings with MAnorm[121] (version 1.1.4). Peaks were assigned to underlying or nearby features and a functional enrichment analysis of these features was performed with ChiP-Enrich[122] (version 3.11). The intersection of ChIP peaks/assigned features with differentially expressed RNA-sequencing results was performed with custom scripts. Normalized coverage was calculated and heatmap/profile plots were made using deepTools[114].

**Reporting summary**. Further information on research design is available in the Nature Research Reporting Summary linked to this article.

## Data availability
The data supporting the findings of this study are available from the corresponding authors upon reasonable request. Source data are provided with this paper. The next-generation sequencing data (DNA-seq, RNA-seq, small RNA-seq and ChIP-seq) generated in this study have been deposited in the NCBI GEO database under accession code GSE172282 (BioProject: PRJNA722747). Analysis of next-generation sequencing data is detailed in the Methods. Custom pipelines executing the software were used, but no custom software or analysis tools were used. The genome version used in this manuscript is *Mus musculus* mm10 from UCSC. All oligonucleotides used in this study are provided in Supplementary Data 5. All *S. pombe* strains generated in this study are

provided in Supplementary Data 4 and are available upon request, as per standard practice. Source data are provided with this paper.

## Code availability
Custom pipelines executing the software were used as described in Methods, using Perl and Bash, but no custom software or analysis tools were used. All custom code is available upon request.

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

## Acknowledgements

We thank Edith Heard (EMBL, Heidelberg, Germany) and Iku Okamoto (Kyoto University) for the inducible *Dicer1* mutant lines, and Chris Vakoc (Cold Spring Harbor Laboratory) for providing advice and reagents and for reading the manuscript. M.J.G. was supported by a Bristol-Myers Squibb Fellowship from the Cold Spring Harbor Laboratory School of Biological Sciences. A.J.S. and B.R. were supported by NIH grant R01GM076396 and R01GM067014 (to R.A.M.). Research in the Martienssen lab is supported by the Howard Hughes Medical Institute (R.A.M.). The authors acknowledge assistance from the Cold Spring Harbor Laboratory Shared Resources, which are funded in part by the Cancer Center Support Grant (5PP30CA045508).

## Author contributions

Conceptualization, M.J.G., B.R., A.J.S., and R.M.; Methodology, M.J.G., K.C., and B.R., Formal Analysis, M.J.G., B.R., and J.I.S.; Investigation, M.J.G., B.R., J.I.S., A.L., and A.J.S. Writing—Original Draft, M.J.G.; Writing—Review & Editing, M.J.G., B.R., and R.M.; Funding Acquisition, R.M.

## Competing interests

The authors declare no competing interests.
