## [Peer Review File · Nature Communications]

Title: Dicer promotes genome stability via the bromodomain transcriptional co-activator BRD4REVIEWER COMMENTS

Reviewer #1 (Remarks to the Author):

Gutbrod et al. examine the possible role of Dicer in nuclear RNAi in mouse embryonic stem cells (mESCs). In addition to its well-established roles in post-translational gene silencing, RNAi is required for some transcriptional gene silencing events and genome stability. These nuclear functions are best understood in fission yeast, plants, and *C. elegans* but whether RNAi also contributes to maintenance of genome stability or transcription regulation in mammalian cells has remained unclear. Some early previous studies observed loss of DNA methylation and centromeric silencing, increased telomere recombination, and chromosome segregation defects in *Dicer*^{-/-} mESCs (Fukagawa et al., 2004, *Nature Cell Biology*; Kanellopoulou et al, 2005, *Genes Dev*) but later studies suggested that these defects were likely indirect and due to loss of specific miRNA(s) that regulate DNA methyltransferase expression (Benetti et al., 2008, *NSMB*; Sinkkonen et al., 2008, *NSMB* – not cited). Gutbrod et al. use a conditional knockout of the RNaseIII domain of Dicer in order to avoid ambiguities that may result from accumulation of suppressor mutation(s) that would allow *Dicer*^{-/-} cells to overcome cell cycle arrest and/or growth defects. They report the following main observations. (1) *Dicer*^{-/-} mESCs have strong proliferation and chromosome segregation defects that induce the interferon response; (2) a CRISPR-cas9 genetic screen to overcome the viability defect of *Dicer*^{-/-} cells uncovers mutation in transcription activators, including the Brd4 transcriptional co-activator and the Elp3 histone acetyltransferase; and (3) mutations in the second bromodomain of Brd4 and its fission yeast homolog rescue *Dicer*-dependent silencing and chromosome segregation defects. Based on these findings they propose that RNAi roles in transcriptional gene silencing of DNA repeats is conserved from fission yeast to mammals. A conserved role for RNAi in these nuclear events would be of great general interest. However, the authors have not ruled that the effects they observe are distinct from previously observed effects that were concluded to be indirect. Some of the findings described by the authors are also inconsistent with this major claim and experiments that would establish direct mechanistic links between different systems are lacking.

Specific comments

1. One of my main concerns is that the authors have not ruled out that the effects they observe following conditional *Dicer* deletion are indirect. Previous studies suggest that the heterochromatin defects of *Dicer*^{-/-} cells were due to loss of the miRNA pathway. One of the arguments the authors use against the possibility that the miRNA pathway is responsible for the effects they observe is based on the differences in % lagging/bridging chromosomes and micronuclei between *Dicer*^{-/-} day 5 KO or clones (propagated for some time) and *Dcgr8*^{-/-} (*Drosha*) clones. The authors make an important point about *Dicer*^{-/-} cells may accumulate suppressors. This may also apply to *Dcgr8*^{-/-} mESCs. They should perform a conditional KO of *Dcgr8* similar to what they have done for *Dicer* to rule out a role for the miRNA pathway. Their results also do not explain the previous observation that the proliferation and DNA methylation defects of *Dicer*^{-/-} cells could be rescued by transfection with miR-290 cluster miRNA (Benetti et al., 2008, *NSMB*; Sinkkonen et al., 2008, *NSMB*).

2. The small RNA analysis data (Figure S3C) is inconsistent with a conserved mechanism of RNAi in both

fission yeast and mammals proposed by the authors. They do not detect any Dicer-dependent small RNAs that map to major satellite repeats as a model based on conservation would suggest. In contrast, they see accumulation of small RNAs which are Dicer-independent and likely represent degradation products of derepressed major satellite repeat RNAs. These data are more consistent with an indirect effect due to loss of DNA methylation or a non-canonical role for Dicer. The authors should test their conditional Dicer^{-/-} cells have lower levels of DNA methylation. In addition, the authors should test whether the proliferation and chromosome segregation defects of Dicer^{-/-} can be rescued by transfection with WT Dicer versus RNaseIII domain 1 and 2 catalytically dead Dicers.

3. The recovery of Brd4 and Elp3 mutations as suppressors of Dicer^{-/-} is a nice result. However, the data regarding major satellite repeats as a direct target of Dicer and Brd4 is only suggestive. It is equally possible that the effect is indirect or mediated through the other 97 genic targets that show increased expression in Dicer^{-/-} and are suppressed by Brd4⁻ (Figure S8E and Figure 4). The data supporting the idea that the reduction in viability of Dicer^{-/-} mESCs is in part due to the accumulation of dsDNA in the cytoplasm (Fig. 3L) is interesting and consistent with chromosome segregation defects and micronuclei accumulation in Dicer^{-/-} cells. Unfortunately, the later may still be indirect effects due to downregulation of DNMTs in Dicer^{-/-} cells.

Reviewer #2 (Remarks to the Author):

This study by Gutbrod and colleagues revisits Dicer knockout ES cells and rescue clones to describe clear suppressors, presenting strong associations uncovering the molecular basis for the phenotypic defects observed and suggests a conserved mode of action for Dicer in silencing centromeric repeats. This is an important finding, which will have a significant impact on the field. Overall, it is a very well implemented, controlled and presented study. For these reasons, in my view the manuscript warrants publication, providing my comments below are addressed.

However, in my view, the story lacks a little in coherence including a number of implied mechanistic links between the molecular activities of the identified suppressors and thus does not provide significant new insights into mechanism. I would suggest that addressing the points below will strengthen the impact of this study, and provide a firmer basis for many of the suggested causal relationships alluded to in the discussion.

Major:

There is an apparent inconsistency between Figures 2 and 3. If the Crd4 BD1 domain is essential as suggested by Figure 2, why doesn't the Brd4 siRNA or JQ1 have proliferation defects in wt cells (Figure S7) and thus no suppressive function in Dicer knockouts (Figure 3) as for the BD1 mutations (Figure 2B)? The Minor satellites are, as frequently the case, left in the shadow of the Major satellites. For example, Figure S8C is not referenced at all in the text. This Figure shows that Minor satellites are similarly misregulated as Major Satellites in Dicer knockouts and their transcription is also rescued by Brd4 and Elp3 mutations. Is strand specific transcription of Minor satellites also affected in a similar way as Major

Satellites? Is Brd4 enriched on Minor Satellites in Dicer Knockouts?

The authors suggest that the partial rescue of MajSat transcription by Brd4 and Elp3 mutations is important for the suppressive activity of these mutants. However only associations are provided as evidence here, and thus 'Dicer1-/- Chromosomal Defects Depend on Major Satellite Transcription' in title of this section should be toned down. As the authors discuss, the altered MajSat transcription could be merely a consequence of cell cycle differences and irrelevant to the phenotypic outcomes, particularly because Brd4 and Elp3 have global effects on transcription (97 likely direct Brd4 direct targets in this context). To provide convincing evidence that the Dicer phenotype is dependent on MajSat transcription, they should directly downregulate MajSat reverse strand transcription, using a TALE or dCas9 fused to KRAB for example. For coherence, it would also be interesting to determine whether this affects Cohesin loading, which itself rescues proliferation (Figure 3I) (see next comment). Further, the reversal of transcription orientation of Major Satellites is intriguing, but again only correlated with phenotypic outcomes. Is the reported effect of ectopic expression of satellite transcripts on chromosome mis-segregation, strand specific? Are forward or reverse MajSat transcripts differentially associated with chromatin, and/or formation of DNA:RNA hybrids? Can they artificially switch major satellite transcription orientation to the forward strand in Dicer knockouts (using a similar dCas9/TALE strategy) and demonstrate a rescue of the phenotypes?

While the identification of three categories of Dicer KO suppressors (transcription, H3K9 methylation and chromosome segregation) and their seemingly consistent associations is an important result, the mechanistic and functional link between them is under-investigated. Which, if any one of these activities, is the downstream mediator of phenotypic defects in Dicer knockouts? More evidence should be provided for the molecular associations between them, to determine causal relationships. For example, RAD21 binding at centromeres is not altered in Dicer single KOs compared to WT cells (S9C). How then can Cohesin binding at centromeres be responsible for the phenotypic defects observed in Dicer single KOs, as alluded to by the authors? Another implication is that the Suv39h1 and Ehmt1 suppressive effects occur through Cohesin. Is RAD21 loading at pericentromeric regions decreased in Suv39h1/Dicer double mutants? What about Major and Minor Satellite transcription? Is reverse orientation MajSat transcription also observed in this context?

Minor:

Page 5: 'both inhibition and overexpression of Aurora B' sentence not finished.

Figure S7A is incorrectly cited as S7C

Reviewer #3 (Remarks to the Author):

In the manuscript "Dicer promotes genome stability via the bromo-domain transcriptional co-activator BRD4" the authors address the role of Dicer in heterochromatic silencing. They find that deletion of

Dicer leads to upregulation of major satellite repeats and reduced viability, which can be rescued by deletion of transcription activators such as Brd4. Furthermore, perturbations in transcription by chemical agents or siRNAs increase viability. This genetic interaction is conserved in fission yeast as well. Although the study offers new insights in Dicer role in heterochromatic silencing in mammalian cells, it does not go beyond genetic interactions. The study is missing more mechanistic data showing what actually Dicer does at these repeats and how it prevents Brd4 recruitment.

1) The authors should provide more mechanistic data showing that Dicer is actually involved in silencing of major satellite repeats. The authors should show if Dicer activity is required for silencing and reduced viability. Although, the authors do not detect Dicer dependent small RNAs, it is possible that Dicer activity plays a role in the process. Alternatively, Dicer might directly interact with transcription machinery to regulate transcription in the repeats.

2) The data suggest that Dicer might somehow regulate transcription, but it is unclear if this is the case. The authors should at least determine if Dicer is localized to the repeats and if it interacts with transcription machinery.

3) The authors show that major satellite repeats are upregulated in Dicer deletion cells. This suggests that Dicer is involved in silencing of those repeats. They observe that in Dicer/Brd4 double mutants the transcripts are still heavily upregulated, however, the opposite strand is now being transcribed. To me it is unclear how this switch in transcription happens and how it suppresses interferon response. The authors should provide more data that the transcription from the opposite strand is resulting in increased viability.

Interesting observation is also that in wild type cells both + and - strands from repeats are transcribed, while in Dicer deletion only + strand and in Dicer/Brd4 deletion cells only – strand is transcribed. The authors should comment why Dicer deletion upregulates + strand and silences – strand.

4) The authors claim that “pericentromeric transcripts in Dicer mutants had strong strand specificity, which was reversed in the triple mutant (Fig. 5C)”. Unfortunately, I do not see this in the data. In all the data presented forward strand is expressed to a higher level than reversed.

REVIEWER COMMENTS

Reviewer #1 (Remarks to the Author):

Gutbrod et al. examine the possible role of Dicer in nuclear RNAi in mouse embryonic stem cells (mESCs). In addition to its well-established roles in post-translational gene silencing, RNAi is required for some transcriptional gene silencing events and genome stability. These nuclear functions are best understood in fission yeast, plants, and *C. elegans* but whether RNAi also contributes to maintenance of genome stability or transcription regulation in mammalian cells has remained unclear. Some early previous studies observed loss of DNA methylation and centromeric silencing, increased telomere recombination, and chromosome segregation defects in *Dicer*^{-/-} mESCs (Fukagawa et al., 2004, *Nature Cell Biology*; Kanellopoulou et al, 2005, *Genes Dev*) but later studies suggested that these defects were likely indirect and due to loss of specific miRNA(s) that regulate DNA methyltransferase expression (Benetti et al., 2008, *NSMB*; Sinkkonen et al., 2008, *NSMB* – not cited). Gutbrod et al. use a conditional knockout of the RNaseIII domain of Dicer in order to avoid ambiguities that may result from accumulation of suppressor mutation(s) that would allow *Dicer*^{-/-} cells to overcome cell cycle arrest and/or growth defects. They report the following main observations. (1) *Dicer*^{-/-} mESCs have strong proliferation and chromosome segregation defects that induce the interferon response; (2) a CRISPR-cas9 genetic screen to overcome the viability defect of *Dicer*^{-/-} cells uncovers mutation in transcription activators, including the *Brd4* transcriptional co-activator and the *Elp3* histone acetyltransferase; and (3) mutations in the second bromodomain of *Brd4* and its fission yeast homolog rescue *Dicer*-dependent silencing and chromosome segregation defects. Based on these findings they propose that RNAi roles in transcriptional gene silencing of DNA repeats is conserved from fission yeast to mammals. A conserved role for RNAi in these nuclear events would be of great general interest. However, the authors have not ruled that the effects they observe are distinct from previously observed effects that were concluded to be indirect. Some of the findings described by the authors are also inconsistent with this major claim and experiments that would establish direct mechanistic links between different systems are lacking.

We thank the reviewer for recognizing the broad appeal of our study and for identifying potential inconsistencies that we have addressed below.

1. One of my main concerns is that the authors have not ruled out that the effects they observe following conditional Dicer deletion are indirect. Previous studies suggest that the heterochromatin defects of *Dicer*^{-/-} cells were due to loss of the miRNA pathway. One of the arguments the authors use against the possibility that the miRNA pathway is responsible for the effects they observe is based on the differences in % lagging/bridging chromosomes and micronuclei between *Dicer*^{-/-} day 5 KO or clones (propagated for some time) and *Dcgr8*^{-/-} (*Drosha*) clones. The authors make an important point about *Dicer*^{-/-} cells may accumulate suppressors. This may also apply to *Dcgr8*^{-/-} mESCs. They should perform a conditional KO of

Dgcr8 similar to what they have done for Dicer to rule out a role for the miRNA pathway. Their results also do not explain the previous observation that the proliferation and DNA methylation defects of *Dicer*^{-/-} cells could be rescued by transfection with miR-290 cluster miRNA (Benetti et al., 2008, NSMB; Sinkkonen et al., 2008, NSMB).

1. We agree that the loss of microRNAs has effects on *Dicer1*^{-/-} cells. However, we and others have provided significant genetic evidence that microRNAs are not driving the majority of the phenotypes we observe.
 - The generation of *Dgcr8*^{-/-} cells by Wang et al. 2008 show that the proliferation defects of freshly derived *Dgcr8*^{-/-} cells is very mild and therefore these cells are significantly less likely to accumulate suppressors as the selective pressure is much less than *Dicer1*^{-/-} mESCs. The commercially available cells we used are low passage number and likely to reflect the effects seen in Wang et al. 2008.
 - Wang et al. 2008 also make this conclusion: “In addition, *Dicer1* knockout ES cells seem to have a more profound initial proliferation defect that is overcome over time, presumably due to additional genetic events. By contrast, *Dgcr8* knockout ES cells show a stable and more subtle proliferation defect. These differences in the phenotypes of *Dicer1* and *Dgcr8* knockout ES cells suggest that Dicer has miRNA-independent roles in ES cell function. *Dgcr8* knockout ES cells provide a means to identify these roles.”
 - The major phenotypes we describe – dysregulation of major satellite repeat transcription, chromosome segregation defects, severe proliferation defects, and a significant increase in apoptosis – have never been reported to occur in mESCs that have lost other RNAi components such as Drosha, Dgcr8, or Argonautes in our work or others. We therefore consider these phenotypes to be generally microRNA-independent. However, we acknowledge that microRNA-regulated gene expression changes likely play a modifying role for some of these phenotypes such as the subset of the proliferation defect that derives from the loss of cell cycle-regulating microRNAs, which can explain some of the cell cycle defects observed in *Dicer1*^{-/-} cells and most of the cell cycle defects observed in *Dgcr8*^{-/-} cells.
 - Recently, identical phenotypes of major satellite transcription increase (also strand-specific) and segregation defects were found in *Dicer1*^{-/-} germ cells (Yadav et al. 2020, Nucleic Acids Research) which have much more severe defects compared to *Dgcr8*^{-/-} germ cells (Zimmerman et al. 2014, PLOS One), extending these observations to additional cell types
 - Finally, similar Dicer phenotypes also occur in *S. pombe*, an organism without microRNAs or DNA methylation. While we recognize that microRNA-driven gene expression changes likely provide additional modification to the phenotypes we observe in mESCs, the origin of these transcriptional phenotypes is conserved and microRNA-independent.
2. The mechanism proposed by Benetti et al. 2008, NSMB and Sinkkonen et al. 2008, NSMB is that miR-290 targets and represses *Rbl2*, a factor which represses DNA

methyltransferases. A reduction in DNMTs in *Dicer1*^{-/-} cells is then thought to lead a reduction in DNA methylation at various elements in the genome depending on the study. Additionally, Sinkkonen et al. 2008 show that transfection of miR-290 does show a mild rescue of proliferation defects in *Dicer1*^{-/-} cells. However, a number of molecular studies argue that loss of methylation does not explain the effects that we describe for the following reasons.

- In general, the observation that *Dicer1*^{-/-} cells lose DNA methylation is very inconsistent. While Benetti et al. claim a reduction in global DNA methylation in *Dicer1*^{-/-} based on a reduction of DNA methylation at B1-SINE elements, other groups do not detect a loss of global DNA methylation – Sinkkonen et al. 2008, Murchison et al. 2005, Calabrese et al. 2007, and Ip et al. 2012. We propose, as others have (Ip et al. 2012, PLOS Genetics), that DNA methylation changes occasionally observed in *Dicer1*^{-/-} mESCs are the result of stochastic variation and selection of small numbers of surviving clones due to the high selective pressure potentially of epialleles.
- The rescue of proliferation defects in *Dicer1*^{-/-} in Sinkkonen et al. 2008 with miR-290 transfection is relatively minor (up to ~30% to ~50% of the proliferation rate of wild type cells) compared to many of our observed rescue experiments (siRNA/pharmacological proliferation assays). Furthermore, miR-290 has been shown to promote the G1/S phase transition (Yuan et al. 2017 Cell & Bioscience) which might have an indirect effect on proliferation.
- In our own RNA-seq data we do observe a modest upregulation of *Rbl2* in *Dicer1*^{-/-} conditions relative to wild type. However, we do not observe a consistent subsequent downregulation of the DNA methyltransferases in either *Dicer1*^{-/-} induced or clonal cell lines relative to wild type. In fact, we find very little change in both *Dnmt1* and *Dnmt3b* levels and significant upregulation of *Dnmt3a* in both conditions. The activity of DNMT3A alone is sufficient for normal methylation levels in mESCs (Chen et al. 2003, Mol. Cell. Biol.). The inconsistency in DNMT protein levels in *Dicer1*^{-/-} mESCs has also been observed by others – Ip et al. 2012, PLOS Genetics. Without the downregulation of DNA methyltransferases no global change is expected in *Dicer1*^{-/-} cells. Therefore, this model does not explain the phenotypes we observe.

We thank the reviewer for suggesting this potential alternative model and have included an addition to the introduction to reflect our consideration for other readers:

In *Dicer1*^{-/-} mESCs, widely differing phenotypes have been reported^{26–28} and one explanation might be the accumulation of mutations that allow stalled *Dicer1*^{-/-} cells to proliferate²⁸. While changes in DNA methylation were initially hypothesized to be partially responsible for proliferation defects^{34,35}, follow-up studies have demonstrated little change in DNA methylation levels in *Dicer1*^{-/-} mESCs which do not explain the phenotypes observed³⁶. Genetic suppressors arise in Dicer mutants of fission yeast when they exit the cell cycle and as RNAi becomes essential, resulting in the selection and outgrowth of suppressed strains¹⁸.

2. The small RNA analysis data (Figure S3C) is inconsistent with a conserved mechanism of RNAi in both fission yeast and mammals proposed by the authors. They do not detect any Dicer-dependent small RNAs that map to major satellite repeats as a model based on conservation would suggest. In contrast, they see accumulation of small RNAs which are Dicer-independent and likely represent degradation products of derepressed major satellite repeat RNAs. These data are more consistent with an indirect effect due to loss of DNA methylation or a non-canonical role for Dicer. The authors should test their conditional Dicer^{-/-} cells have lower levels of DNA methylation. In addition, the authors should test whether the proliferation and chromosome segregation defects of Dicer^{-/-} can be rescued by transfection with WT Dicer versus RNaseIII domain 1 and 2 catalytically dead Dicers.

We appreciate the opportunity to more clearly state our proposed mechanism. Our work presented here shows that the majority of the phenotypic effects we describe are independent of small RNAs. As the reviewer points out we make a point of stating that we do not see Dicer-dependent major satellite small RNAs. Thus, the conserved mechanism we propose is not the well-described small RNA-directed chromatin formation mechanism that occurs in yeast and plants (but has not been shown to occur in mammals), but rather a conserved non-canonical role for Dicer that occurs at the transcription of pericentromeric repeats.

Our data indicates that Dicer plays a critical, direct role in the processing of pericentromeric transcription in both mouse and yeast.

- Upon the loss of Dicer, transcription of this region increases abnormally, in a strand specific manner as we and others have observed (Yadav et al. 2020, NAR), this results in failure of cohesion, chromosome segregation defects, cytosolic DNA, and an increase in interferon signaling all of which reduce proliferation and increase apoptosis. We observe the same effects in *S. pombe* (an organism without microRNAs) and show they can be rescued by targeting the exact same factors that directly regulate pericentromeric transcription in both yeast and mammals.
- As stated above, our model is not consistent with a global loss of DNA methylation and even a local loss of DNA methylation at major satellite repeats has not been observed in *Dicer1*^{-/-} cells (Calabrese et al. 2007, PNAS and Ip et al. 2012, PLOS Genetics).

- The inducible *Dicer1*^{-/-} alleles that we generate are a loss of both RNase III domains. We predict that both RNaseIII domains are likely required for processing substrates associated with these phenotypes. A more detailed examination is beyond the scope of this work.

Lastly, we provide additional data strongly supporting our model that direct regulation of transcription rather than post-transcriptional regulation is critical for generating and suppressing the phenotypes.

- We see no major satellite-derived small RNAs that are lost upon mutating *Dicer1* (Fig. S3C).
- We can suppress the viability and proliferation defects by simply inhibiting RNA Pol II with low doses of alpha-amanitin (Fig. 3C).
- Reducing Argonaute expression does not suppress the viability defects (new Figs. S3 D-G, shown below).

We have also added a sentence to the text to address this new data:

The lack of DICER1-dependent siRNA is consistent with the absence of an RdRP. **Furthermore, the knockdown of all mouse Argonaute proteins simultaneously did not affect the proliferation of *Dicer1*^{-/-} mESCs in a viability assay (Figs. S3 D-G) also demonstrating that small RNA are unlikely to be the primary driver of the *Dicer1*^{-/-} phenotypes.** We also performed ChIP-seq, and observed a modest reduction in H3K9me3 at some ERVs, namely IAP and ETn, as well as at LINE1 transposable element loci (Figs. S4A, S4B, S4D, and S5B), as described previously⁴⁴, which might be related to the loss of microRNA.

3. The recovery of Brd4 and Elp3 mutations as suppressors of Dicer^{-/-} is a nice result. However, the data regarding major satellite repeats as a direct target of Dicer and Brd4 is only suggestive. It is equally possible that the effect is indirect or mediated through the other 97 genic targets that show increased expression in Dicer^{-/-} and are suppressed by Brd4⁻ (Figure S8E and Figure 4).

Our data described here as well as data published by others have shown conclusively that DICER1 and BRD4 directly target the major satellite repeat.

- A direct interaction between the DICER1 protein and the major satellite genomic loci as well as transcripts has been detected by ChIP and RNA pull-down respectively in mouse cells (Hsieh et al. 2011 Nucleic Acids Research).
- Additionally, in mouse spermatogenesis the loss of DICER1 induces major satellite transcription in a strand-specific manner (as we observe) and furthermore, a direct association between the DICER1 protein and the major satellite genomic loci was detected using a combined immunofluorescence/*in situ* hybridization microscopy approach as well as ChIP-PCR (Yadav et al. 2020 Nucleic Acids Research).
- We also observe a significant increase in major satellite transcription (it is in fact the top upregulated transcript) when we ablate DICER1 activity. Therefore, the detection of direct regulation in multiple cell types and increased transcriptional output upon DICER1 loss across three cell types in mouse is convincing evidence that DICER1 directly targets the major satellite repeats.
- Our combined ChIP-seq and RNA-seq datasets demonstrate that BRD4 is clearly bound to the major satellite genomic loci (Fig. 4A) and regulates the transcription in this region (Figs. 4C and S8B)1. Furthermore, we show that the BRD4 ortholog Bdf2 also regulates pericentromeric transcription in *S. pombe*. This protein has been shown to bind pericentromeric repeats and accumulate at centromeric boundaries where the promoters lie (Wang et al. 2013, Gene & Dev)
- Finally, the localization of BRD4 to the pericentromeric chromatin has been demonstrated in human cells as well (Ishikura et al. 2020 Nucleic Acids Research). This paper convincingly shows that direct perturbations of BRD4 activity with JQ1 or siRNAs reduced pericentromeric transcription as we have.
- Thus, we and others have shown that BRD4 is localized to pericentromeric repeats and directly regulates the transcription of these repeats across evolution.

While we propose a direct regulation of the major satellite by DICER1 and BRD4 the other 97 genes that fit the same expression pattern could be playing a role as well and we thank the reviewers for bringing up this point. We have now closely interrogated each of these 97 candidates and passed them through a series of filters as follows.

- Initially, we intersected our RNA-seq dataset with RNA-seq from both *Droscha* knockout mESCs (Georgakilas et al. 2014 Nature Comms) and *Dgcr8* knockout mESCs (Cirera-Salinas et al. 2017 J. Cell Biology) in order to determine which of these genes were microRNA targets and thus not likely to generate these phenotypes as they do not occur in other microRNA-pathway mutants. This removed 34 of the genes.

- Furthermore, as we observed the same phenotypes and the same genetic relationships in *S. pombe* we would expect a true modifying locus to be conserved. We found only 7 genes that were not microRNA targets and were conserved in fission yeast. Of these 7, none of them were misregulated in Dicer mutants in *S. pombe* (Hansen et al. 2005, MCB).
- In parallel, we did a deep literature search of all 97 genes to identify any that had been associated with chromosome segregation and we found that none had direct mechanistic links to this phenotype.
- We did find that one of these 97 genes - *Prdm16*, an H3K9me1 methyltransferase - does act at the centromeric heterochromatin, but is functionally redundant with *Prdm3* (Pinheiro et al. 2012, Cell), which is not differentially expressed in our datasets and therefore *Prdm16* is not a candidate to drive these phenotypes.
- Therefore, none of the other candidate genes had a strong link to chromosomal defects and this led us to focus on the pericentromeric satellite repeats themselves, as increased transcription has been shown to lead to chromosomal abnormalities in mouse cells (Zhu et al. 2018, Molecular Cell) and these transcripts showed by far the greatest magnitude of misregulation in *Dicer1*^{-/-} cells while this misregulation was also rescued by the *Brd4* suppressors.

We have added to the main text to further detail our treatment of the 97 candidates:

97 genes were upregulated upon *Dicer1* mutation, downregulated upon *Brd4* and *Elp3* mutation, and were located near a BRD4 ChIP-seq peak, suggesting they were direct targets of both DICER1 and BRD4. We closely examined all 97 of these candidates through – (i). Cross-referencing *Dgcr8* and *Drosha* knockout mESC RNA-seq datasets to eliminate microRNA target genes, (ii). Searching for homologs in *S. pombe* as Dicer-dependent centromeric silencing is evolutionarily conserved, and (iii). A deep literature search to identify roles in chromosome segregation that could be generating this critical phenotype. None of the protein-coding candidates we examined passed these filters.

In contrast, the major satellite transcript was the most upregulated transcript in *Dicer1*^{-/-} cells, and activation of major satellite transcription has been shown to cause chromosome segregation defects in mouse cells⁵⁶.

The data supporting the idea that the reduction in viability of *Dicer*^{-/-} mESCs is in part due to the accumulation of dsDNA in the cytoplasm (Fig. 3L) is interesting and consistent with chromosome segregation defects and micronuclei accumulation in *Dicer*^{-/-} cells. Unfortunately, the later may still be indirect effects due to downregulation of DNMTs in *Dicer*^{-/-} cells.

We appreciate the reviewer recognizing the consistency of phenotypes. As we described in our response above, our RNA-seq data showed we do not observe a downregulation of DNMTs upon the loss of *Dicer1*. Therefore, we propose the segregation defects leading to dsDNA in the cytoplasm is a direct effect of DICER1 acting on transcription of the pericentromeric repeat.

Reviewer #2 (Remarks to the Author):

This study by Gutbrod and colleagues revisits Dicer knockout ES cells and rescue clones to describe clear suppressors, presenting strong associations uncovering the molecular basis for the phenotypic defects observed and suggests a conserved mode of action for Dicer in silencing centromeric repeats. This is an important finding, which will have a significant impact on the field. Overall, it is a very well implemented, controlled and presented study. For these reasons, in my view the manuscript warrants publication, providing my comments below are addressed.

However, in my view, the story lacks a little in coherence including a number of implied mechanistic links between the molecular activities of the identified suppressors and thus does not provide significant new insights into mechanism. I would suggest that addressing the points below will strengthen the impact of this study, and provide a firmer basis for many of the suggested causal relationships alluded to in the discussion.

We thank the reviewer for recognizing the importance of this study to the field and we hope our response will sufficiently address the concerns regarding a lack of coherence between mechanistic links.

There is an apparent inconsistency between Figures 2 and 3. If the Brd4 BD1 domain is essential as suggested by Figure 2, why doesn't the Brd4 siRNA or JQ1 have proliferation defects in wt cells (Figure S7) and thus no suppressive function in Dicer knockouts (Figure 3) as for the BD1 mutations (Figure 2B)?

We appreciate the reviewer pointing out the seemingly conflicting data on bromodomain 1.

- We have found that in wild type mESCs CRISPR targeting of BD1 does lead to more deleterious effects than targeting BD2 – cells containing BD1-targeting guides drop out of the population more readily than BD2-targeting guides and this effect continues progressively over time (Figure 2B).
- Despite these negative effects, we find that from days 8-12 of the *Dicer1*^{-/-} timecourse targeting BD1 does suppress the *Dicer1*^{-/-} proliferation defects, but deeper into the timecourse (between 12-16 days) this effect is overcome by the negative consequences of specifically targeting BD1 (Figure 2B).
- While the insertions/deletions generated in BD1 by Cas9 are deleterious, the inhibition of BRD4 by JQ1, which binds both BD1 and BD2, or siRNA do not inhibit proliferation or viability over the 2-3 day timeframe of the luciferase-based MT assays (Figure 3). We hypothesize that it is the different targeting modality (permanent indels vs transitory inhibition) and different time frame that underly this apparent discrepancy. As observed by others, this shows that BD1 clearly has more essential functions in the cell than BD2, which does in fact tolerate loss-of-function mutations. Additionally in *S. pombe*, the

bdf1 and bdf2 double mutant is synthetic lethal and the domain responsible for the synthetic lethality is the BD1 of bdf2 as shown by our tetrad analysis.

- Importantly, we see the opposite relationship with *Elp3* in *Dicer1*^{-/-} cells. The Cas9 targeted mutations in the histone methyltransferase domain are excellent suppressors of the *Dicer1* proliferation and viability defects, but siRNA treatment (at multiple concentrations) does not show suppression in the MT viability assays.
- Therefore, it is critical to use multiple mechanisms of perturbation that have differing effects over time in order to more completely understand a gene's relationship to proliferation.

We have added further explanation to the text to address this point:

Most significantly, the *Dicer1*^{-/-} viability defect was strongly rescued by the small molecule inhibitor JQ1, which specifically inhibits BRD4 and its paralogs (Fig. 3D). While targeting BRD4 with siRNAs or JQ1 does inhibit both bromodomains, we do not observe the deleterious effects of targeting BD1 with these modalities in either *Dicer1*^{-/-} or wild type mESCs (Figs. 3D, 3E, S7B, and S7C). This is likely due to the transitory and less efficient nature of these treatments in comparison to generating frameshift mutations with CRISPR as in Figure 2.

The Minor satellites are, as frequently the case, left in the shadow of the Major satellites. For example, Figure S8C is not referenced at all in the text. This Figure shows that Minor satellites are similarly misregulated as Major Satellites in Dicer knockouts and their transcription is also rescued by Brd4 and Elp3 mutations. Is strand specific transcription of Minor satellites also affected in a similar way as Major Satellites? Is Brd4 enriched on Minor Satellites in Dicer Knockouts?

We appreciate the reviewer pointing this out. We have now also examined the regulation of the minor satellite repeat transcripts extensively.

- We did see a similar pattern of expression level change, but the relative expression and the fold change was much lower. In our RNA-seq data we again saw a similar pattern, but the effect was orders of magnitude greater for the major satellite transcripts as we saw in our RT-qPCR (see data below and new figure Fig S8D).
- The very low read count (<10 in some samples) for the minor satellite prevented us from determining strand specificity in our RNA-seq data.
- We also did not see BRD4 enrichment at the minor satellite consensus in our ChIP-seq and ChIP-qPCR data.

We have incorporated this additional figure and have added to the text to highlight these results:

In contrast, the major satellite transcript was the most upregulated transcript in *Dicer1*^{-/-} cells, and activation of major satellite transcription has been shown to cause chromosome segregation defects in mouse cells⁵⁶. **The minor satellite transcript on the other hand was very lowly expressed and relatively stable in all conditions (Figs. S8C and S8D).** The abundance of the major satellite transcripts increased dramatically over the culture time-course, but was reduced in viable clones and in *Dicer1*^{-/-} d8 cells with *Brd4* or *Elp3* mutations (Fig. 4C).

The authors suggest that the partial rescue of MajSat transcription by *Brd4* and *Elp3* mutations is important for the suppressive activity of these mutants. However only associations are provided as evidence here, and thus 'Dicer1^{-/-} Chromosomal Defects Depend on Major Satellite Transcription' in title of this section should be toned down. As the authors discuss, the altered MajSat transcription could be merely a consequence of cell cycle differences and irrelevant to the phenotypic outcomes, particularly because *Brd4* and *Elp3* have global effects on transcription (97 likely direct *Brd4* direct targets in this context). To provide convincing evidence that the *Dicer* phenotype is dependent on MajSat transcription, they should directly downregulate MajSat reverse strand transcription, using a TALE or dCas9 fused to KRAB for example.

We have edited the title to reflect the reviewer's comments:

This is likely due to the transitory and less efficient nature of these treatments in comparison to generating frameshift mutations with CRISPR as in Figure 2.

***Dicer1*^{-/-} Chromosomal Defects are Associated with Major Satellite Transcription**

In order to investigate the mechanism of suppression, we performed BRD4 ChIP-seq in wild type and *Dicer1*^{-/-} mESCs.

As discussed above (in response to Reviewer 1) we have now closely examined all 97 of these genes and found the major satellite repeat to be by far the most likely locus generating these phenotypes.

We agree that a direct perturbation of major satellite transcription would provide convincing evidence for its role in chromosome segregation. We have now attempted downregulation of the major satellite transcripts with ASOs at many concentrations spanning 5 orders of magnitude and targeting each strand individually as well as combined, but have not observed any amelioration of the viability/proliferation defects in *Dicer1*^{-/-} mESCs (below, new Fig S7L-M) potentially due to the massive magnitude of upregulation that we observe upon the loss of *Dicer1* (Figs 1D, 4C, S8B). However, another group has shown that CRISPR activation of major satellite transcription in mouse cells does lead to chromosomal defects exactly as we describe (Zhu et al. 2018, Molecular Cell). This effect is conserved in humans and promotes cancer. Combined, these experiments indicate that it is the act of transcription rather than the transcripts themselves that is driving the phenotype. Most convincingly, we identify transcriptional activators that directly bind the major satellite loci (BRD4) as the strongest suppressors of the viability/proliferation defects as well as the chromosomal segregation defects in both mouse and yeast cells. These results strongly support our identification of major satellite transcription as the driver of the chromosomal phenotypes.

We have added these figures and updated the text:

This transcript was also strongly downregulated in transcriptomes from *Brd4*^{BD2^{-/-}} *Dicer1*^{-/-} and *Elp3*^{HAT^{+/+}} *Dicer1*^{-/-} clonal double mutants (Fig. S8B). Targeting of the major satellite transcripts with antisense oligonucleotides (ASOs) did not suppress the viability and proliferation defects of *Dicer1*^{-/-} mESCs across a range of concentrations or with strand-specific ASOs tested either individually or in combination (Figs. S7L and S7M). However, the mutation of *Brd4* strikingly reversed strand-specific transcription of the satellite transcripts in cultured *Dicer1*^{-/-} cells, closely resembling transcription in viable clones that had undergone the strong selection (Fig. 4B). Along with reduced BRD4 occupancy at satellite loci (Fig. S9A), and a reduction in elongating Pol II (Fig. S9B), chromosomal defects of *Dicer1*^{-/-} cells were also rescued by *Brd4*^{BD2^{-/-}} and *Elp3*^{HAT^{+/+}} (Fig. 4D). We further detected a substantial reduction of RAD21 at both the major and minor satellite loci in *Brd4*^{BD2^{-/-}} *Dicer1*^{-/-} double mutants (Fig. S9C) consistent with recent

findings that BRD4 interacts directly with RAD21 in human cells⁵⁷ and in *D. melanogaster*⁵⁸ as well as with NIPBL in humans^{57,59}.

While we appreciate the suggestion of the reviewer to use CRISPRi dCas9-KRAB we found there to be a number of issues with this strategy.

- KRAB domains induce H3K9me3-dependent silencing, and we have found that H3K9me2/3 is increased in *Dicer1*^{-/-} cells, rather than decreased (Fig S4C). Therefore, we predict that targeting this region with a KRAB domain may not induce further increases and transcriptional silencing.
- This strategy generates heterochromatin indiscriminately of strand and will not allow us to test the strand-specific effect we observe as the reviewer suggests.
- The hundreds of thousands of loci (though only a fraction of these are transcriptionally active) represent a major challenge for the CRISPRi system. The design of sgRNAs that target all of the sequence variations of major satellite transcripts and use the standard NGG PAM sequence in the AT-rich major satellite has been a major challenge, especially as the promoter region is unknown (Zhu et al, 2018). The level of reduction we observe in *Brd4* suppressor mutants may be difficult to mimic with this strategy.

For coherence, it would also be interesting to determine whether this affects Cohesin loading, which itself rescues proliferation (Figure 3I) (see next comment). Further, the reversal of transcription orientation of Major Satellites is intriguing, but again only correlated with phenotypic outcomes. Is the reported effect of ectopic expression of satellite transcripts on chromosome mis-segregation, strand specific? Are forward or reverse MajSat transcripts differentially associated with chromatin, and/or formation of DNA:RNA hybrids? Can they artificially switch major satellite transcription orientation to the forward strand in *Dicer* knockouts (using a similar dCas9/TALE strategy) and demonstrate a rescue of the phenotypes?

As detailed above we have tried extensively to use strand-specific perturbations such as ASOs to suppress the phenotype, but have been unable to demonstrate an effect on the viability and proliferation defects. While it has been clearly demonstrated that major satellite transcripts can associate with the chromatin and heterochromatin factors such as SUV39H1 (Comacho et al. 2017, eLife and Johnson et al. 2017, eLife) we observe very little change in heterochromatin state despite the massive upregulation of major satellite transcripts. This has led us to propose that it is the act of transcription rather than the transcripts themselves that is driving the phenotype. Thus, we can easily suppress the phenotypes by targeting transcription factors such as *Brd4*.

We have also considered a model in which DNA:RNA hybrids play a role. However, it has recently been demonstrated that BRD4 actually prevents the accumulation of R-loops (Lam et al. 2020 Nature Comms, Edwards et al. 2020 Cell Rep.). Therefore, it is unlikely that the primary mechanism generating these phenotypes is related to DNA:RNA hybrids.

We have added to the text to reflect this consideration:

We considered the possibility that the major satellite transcripts generate DNA:RNA hybrids or R-loops that lead to deleterious effects in *Dicer1*^{-/-} mESCs. However, our strongest suppressor, *Brd4*, has been shown to prevent the accumulation of R-loops^{80,81} and so mutations would likely only further increase the accumulation of R-loops in *Dicer1*^{-/-} cells. In the absence of RdRP, and consistent with the lack of small RNAs, there was no decrease in H3K9me2/3 at the pericentromere, while there was a decrease of H3K9me3 at retrotransposons that have corresponding small RNAs⁴⁴.

Finally, as mentioned above the CRISPRi dCas9-KRAB/TALE strategies would not generate strand-specific disruption of major satellite transcription and would not allow us to test this hypothesis. We conclude that mutation of transcription factors such as *Brd4* and *Elp3* may be the only experimental paradigm powerful enough to combat the high levels of major satellite transcription in a strand-specific way.

While the identification of three categories of *Dicer* KO suppressors (transcription, H3K9 methylation and chromosome segregation) and their seemingly consistent associations is an important result, the mechanistic and functional link between them is under-investigated. Which, if any one of these activities, is the downstream mediator of phenotypic defects in *Dicer* knockouts? More evidence should be provided for the molecular associations between them, to determine causal relationships. For example, RAD21 binding at centromeres is not altered in *Dicer* single KOs compared to WT cells (S9C). How then can Cohesin binding at centromeres be responsible for the phenotypic defects observed in *Dicer* single KOs, as alluded to by the authors? Another implication is that the *Suv39h1* and *Ehmt1* suppressive effects occur through Cohesin. Is RAD21 loading at pericentromeric regions decreased in *Suv39h1/Dicer* double mutants? What about Major and Minor Satellite transcription? Is reverse orientation MajSat transcription also observed in this context?

We appreciate that additional clarification and interpretation of our results would help readers understand our model. While we do identify H3K9 methyltransferases as suppressors the K9me2/3 ChIP-seq and immunofluorescence experiments (Figs. S4 and S5) do not change significantly. This leads us to conclude that transcription is the critical mechanism underlying these phenotypes. To avoid being too vague in our interpretations, we have altered the text to reflect this.

In ES cells, we have shown that the *Dicer1*^{-/-} viability defect is due to transcription of the centromeric satellite repeats, and can be rescued by hypomorphic mutations in transcription factors *Brd4* and *Elp3* or by inhibiting Pol II. In support of our findings, nuclear-localized DICER1 has been found associated with satellite repeats and their transcripts, and with the nuclear protein WDHD1 in complex with Pol II²⁹ and regulates transcription at this locus in multiple mouse cell types^{29,73}. *Dicer* has been reported to interact with Pol II in *Drosophila*⁷⁴ and in human HEK293 cells, where it also prevents the accumulation of dsRNAs from satellite

repeats⁷⁵. Additionally, there are well-described interactions between Dicer and the transcriptional machinery in *S. pombe*^{12,76,77}.

The fact that *Brd4* and *Elp3* were the strongest suppressors of the *Dicer1*^{-/-} phenotype, which was also suppressed with low doses of α -amanitin, strongly implies transcription as the underlying mechanism. We found strand-specific accumulation of major satellite transcripts in *Dicer1*^{-/-} mESCs that was reversed by *Brd4*^{BD2}^{-/-} mutations and by selection for viability that generated our clonal lines.

In fact, we find the clearest associations between classes of suppressors occur with transcription factors and chromosome segregation factors. While RAD21 does not accumulate in *Dicer1* mutants as the reviewer points out, the mutation of *Brd4* does significantly reduce RAD21 occupancy at the major satellite loci and overall reduction of RAD21 levels with siRNAs suppresses the viability and proliferation defects of the *Dicer1*^{-/-} mESCs (Fig. 3I). In fact, a missense mutation in the second bromodomain of BRD4 has been shown to cause a version of the cohesinopathy (Cornelia DeLange Syndrome) through the loss of cohesin loading by NIPBL in the BRD4 mutant (Olley et al. 2018 Nat. Genetics). Again, we propose it is not necessary for RAD21/Cohesin to over-accumulate and be reduced by targeting BRD4 or RAD21 directly, but rather that the defects caused by the loss of DICER1 at this region can be ameliorated by reducing Cohesin at the pericentromere.

While it is possible RAD21 loading is reduced by targeting H3K9 methyltransferases as well, we have not experimentally tested this hypothesis and we felt this investigation lies outside of the scope of this study. We found a much stronger effect in suppressing with transcription factors like BRD4 and decided to focus on this aspect.

Minor:

Page 5: 'both inhibition and overexpression of Aurora B' sentence not finished.
Figure S7A is incorrectly cited as S7C

We thank the reviewer for bringing our attention to these editorial errors and they have been corrected.

Reviewer #3 (Remarks to the Author):

In the manuscript "Dicer promotes genome stability via the bromo-domain transcriptional co-activator BRD4" the authors address the role of Dicer in heterochromatic silencing. They find that deletion of Dicer leads to upregulation of major satellite repeats and reduced viability, which can be rescued by deletion of transcription activators such as Brd4. Furthermore, perturbations in transcription by chemical agents or siRNAs increase viability. This genetic interaction is conserved in fission yeast as well. Although the study offers new insights in Dicer role in heterochromatic silencing in mammalian cells, it does not go beyond genetic

interactions. The study is missing more mechanistic data showing what actually Dicer does at these repeats and how it prevents Brd4 recruitment.

We thank the reviewer for their comments and hope we have demonstrated that we have indeed gone beyond only describing genetic interactions in this study.

1) The authors should provide more mechanistic data showing that Dicer is actually involved in silencing of major satellite repeats. The authors should show if Dicer activity is required for silencing and reduced viability. Although, the authors do not detect Dicer dependent small RNAs, it is possible that Dicer activity plays a role in the process. Alternatively, Dicer might directly interact with transcription machinery to regulate transcription in the repeats.

Our model is for a non-canonical role of Dicer in regulating the transcription of the major satellite repeats rather than the well-described small RNA-dependent silencing mechanism. As the reviewer points out we do not detect Dicer-dependent small RNAs and we have now additionally found that targeting Argonautes does not suppress the viability and proliferation defects (see response to Reviewer 1's point #2). We have clearly shown that Dicer is involved in directly regulating the transcription of pericentromeric repeats in both mouse and yeast. We find that the loss of Dicer leads to massive upregulation of pericentromeric repeat transcripts in both organisms (Figs. 1D, 4C, S8B, 5C). We have also demonstrated that the loss of Dicer leads to reduced viability in both organisms (Figs. S1B, 3A, 5A, and 5B). Additionally, the fact that mutations in the same transcription factors that directly bind and control transcription of the pericentromeric repeats suppress so strongly in two evolutionarily distant organisms is suggestive of a conserved mechanism. Finally, we have pointed to other studies supporting this direct regulation of major satellite transcripts across cell types in mouse:

- A direct interaction between the DICER1 protein and the major satellite genomic loci as well as transcripts has been detected by ChIP and RNA pull-down respectively in mouse cells (Hsieh et al. 2011 Nucleic Acids Research).
- Additionally, in mouse spermatogenesis the loss of DICER1 induces major satellite transcription in a strand-specific manner (as we observe) and furthermore, a direct association between the DICER1 protein and the major satellite genomic loci was detected using a combined immunofluorescence/*in situ* hybridization microscopy strategy as well as ChIP-PCR (Yadav et al. 2020 Nucleic Acids Research).

Nuclear interaction between DICER1 and RNA Pol II has been reported through the WDHD1 protein (Hsieh et al. 2011 Nucleic Acids Research). Additionally, there are well-known interactions between Dicer and transcription in *S. pombe* (Djupedal et al. 2007, Genes & Dev, Reyes-Turcu et al. 2011, NSMB, and Zaratiegui et al. 2011, Nature) We have updated the text to reflect this.

In ES cells, we have shown that the *Dicer1*^{-/-} viability defect is due to transcription of the centromeric satellite repeats, and can be rescued by hypomorphic mutations in transcription

factors *Brd4* and *Elp3* or by inhibiting Pol II. In support of our findings, nuclear-localized DICER1 has been found associated with satellite repeats and their transcripts, and with the nuclear protein WDHD1 in complex with Pol II²⁹ and regulates transcription at this locus in multiple mouse cell types^{29,73}. Dicer has been reported to interact with Pol II in *Drosophila*⁷⁴ and in human HEK293 cells, where it also prevents the accumulation of dsRNAs from satellite repeats⁷⁵. Additionally, there are well-described interactions between Dicer and the transcriptional machinery in *S. pombe*^{12,76,77}.

2) The data suggest that Dicer might somehow regulate transcription, but it is unclear if this is the case. The authors should at least determine if Dicer is localized to the repeats and if it interacts with transcription machinery.

As detailed above other groups have shown across mouse cell types a direct interaction between the DICER1 protein and both the major satellite genomic loci by ChIP and the RNA transcripts and that a loss of DICER1 at these loci leads to upregulation of major satellite transcripts (as we observe).

3) The authors show that major satellite repeats are upregulated in Dicer deletion cells. This suggests that Dicer is involved in silencing of those repeats. They observe that in *Dicer/Brd4* double mutants the transcripts are still heavily upregulated, however, the opposite strand is now being transcribed. To me it is unclear how this switch in transcription happens and how it suppresses interferon response. The authors should provide more data that the transcription from the opposite strand is resulting in increased viability.

Interesting observation is also that in wild type cells both + and - strands from repeats are transcribed, while in *Dicer* deletion only + strand and in *Dicer/Brd4* deletion cells only – strand is transcribed. The authors should comment why *Dicer* deletion upregulates + strand and silences – strand.

We observe a massive upregulation of the major satellite transcripts in *Dicer1* mutants and a very significant reduction in major satellite transcripts in the *Dicer1/Brd4* double mutants, though as the reviewer points out this is still upregulated relative to wild types. We did note the strand-specific trend that appears in the *Dicer1* *-/-* escaping clones and double mutants with *Brd4*, but we feel a full description of the mechanism of strand specificity is a subject for a follow up study. We have attempted to repress the major satellite transcripts in a strand-specific manner with ASOs, but this had not led to an improvement in viability despite extensive variation across a range of concentrations and target sites (see data in response to Reviewer 2's point 3).

We would like to point out that our data does not suggest that only the + strand is being transcribed in *Dicer1* *-/-* cells or that only the – strand is being transcribed in suppressed cells, but rather that there is an enrichment for these strands. The data being plotted in Fig. 4B is a log₂ transformation of the ratio of forward to reverse. We found upregulation of both

transcripts in *Dicer1*^{-/-} conditions, but there is a clear strand bias in each genotype as we present. We have updated the text to include a comment about this to make it more clear for the reader.

However, the mutation of *Brd4* strikingly reversed strand-specific transcription of the satellite transcripts in cultured *Dicer1*^{-/-} cells, closely resembling transcription in viable clones that had undergone the strong selection (Fig. 4B). **In this figure the data being plotted is a log₂ transformation of the ratio of the abundance of the forward strand to the reverse strand.** Along with reduced BRD4 occupancy at satellite loci (Fig. S9A), and a reduction in elongating Pol II (Fig. S9B), chromosomal defects of *Dicer1*^{-/-} cells were also rescued by *Brd4*^{BD2^{-/-}} and *Elp3*^{HAT +/-} (Fig. 4D).

4) The authors claim that “pericentromeric transcripts in *Dicer* mutants had strong strand specificity, which was reversed in the triple mutant (Fig. 5C)”. Unfortunately, I do not see this in the data. In all the data presented forward strand is expressed to a higher level than reversed.

We thank the reviewer for pointing out this phrasing error in our interpretation of the data. We meant to say that the ratio of forward/reverse is much closer to equal in the triple mutant and therefore the strand specificity is reduced. We have updated the text to address this point.

As in mammalian cells, pericentromeric transcripts in *Dicer* mutants displayed strong strand-specificity. **However, in the triple mutant we found that the ratio of the forward strand to the reverse strand was much closer to equal (Fig. 5C), indicating strand specificity is reversed in this strain.** These results indicate that Dcr1 centromeric function is deeply conserved in mESCs and fission yeast, both in cycling cells and upon G₀ entry, and distinguishes between bromodomains BD1 (essential function) and BD2 (silencing function) of the transcription factor BRD4.

REVIEWERS' COMMENTS

Reviewer #1 (Remarks to the Author):

The authors provide reasonable and convincing responses to my concerns based on the results in Wang et al. 2008 and other reports and their own finding arguing that some Dicer^{-/-} phenotypes are independent of the miRNA pathway. The authors similarly provide convincing arguments and review of past findings regarding inconsistencies in loss of DNA methylation in Dicer^{-/-} mESCs. They have added a sentence in the intro with regards to past studies and I am satisfied with their response. I am also satisfied with their response regarding possible direct roles of Dicer and Brd4 in regulation of major satellite repeats, and their clarification that their model proposes a non-canonical small RNA-independent role for Dicer in regulation of transcription.

The authors still conclude that “Dicer-dependent centromeric silencing is evolutionarily conserved.” However, their study shows that the silencing of MSRs by Dicer occurs independently of small RNAs or AGO proteins (an interesting novel contribution of their study). In contrast, in *S. pombe* Dicer functions together with an AGO protein and its silencing functions depend on small RNAs and the RNAi machinery. The authors may wish to consider this point and avoid giving the impression that they imply conservation of analogous mechanisms.

The findings are exciting and of broad interest. I support publication.

Reviewer #2 (Remarks to the Author):

The authors' efforts to address my comments are appreciated, both in providing additional experiments and textual changes in the manuscript. I accept the authors' responses, with one exception. I appreciate the authors' attempts to directly regulate major satellite transcription and while I agree with their interpretation that the data is highly suggestive of a function for major satellite transcription in Dicer mutant phenotypes, this is not directly demonstrated. For the most part the text is consistent with this interpretation. However in the discussion, they argue: 'In ES cells, we have shown that the Dicer1^{-/-} viability defect is due to transcription of the centromeric satellite repeats'. I think this sentence needs to be toned down before publication, in keeping with the rest of the revised manuscript.

Reviewer #3 (Remarks to the Author):

In the revised version the authors have addressed all my concerns.

Response to Reviewers (final version)

Reviewer #1 (Remarks to the Author):

The authors provide reasonable and convincing responses to my concerns based on the results in Wang et al. 2008 and other reports and their own finding arguing that some Dicer^{-/-} phenotypes are independent of the miRNA pathway. The authors similarly provide convincing arguments and review of past findings regarding inconsistencies in loss of DNA methylation in Dicer^{-/-} mESCs. They have added a sentence in the intro with regards to past studies and I am satisfied with their response. I am also satisfied with their response regarding possible direct roles of Dicer and Brd4 in regulation of major satellite repeats, and their clarification that their model proposes a non-canonical small RNA-independent role for Dicer in regulation of transcription.

The authors still conclude that “Dicer-dependent centromeric silencing is evolutionarily conserved.” However, their study shows that the silencing of MSR by Dicer occurs independently of small RNAs or AGO proteins (an interesting novel contribution of their study). In contrast, in *S. pombe* Dicer functions together with an AGO protein and its silencing functions depend on small RNAs and the RNAi machinery. The authors may wish to consider this point and avoid giving the impression that they imply conservation of analogous mechanisms.

The findings are exciting and of broad interest. I support publication.

We would like to thank the reviewer for their comments, and appreciate their helpful suggestions for improving this manuscript.

We agree with the reviewer that a major difference between *S. pombe* centromeric silencing and this novel centromeric function of Dicer in ES cells is indeed in regards to the involvement of Argonaute and small RNAs, suggesting that while a role for Dicer at the centromere is evolutionarily conserved, it may rely on a non-canonical function of Dicer. In this regard, it is interesting to note that some novel functions of Dicer in *S. pombe* are also found to not rely on small RNAs (such as G0 viability in ref. 18, Roche et al.) or not rely on Argonaute (such as rDNA CNV in ref. 13, Castel et al.). To reflect this observation in the text and make it clearer, we have added the following sentence:

“The different role played by small RNA and Argonaute proteins in these two model organisms may reflect distinct mechanistic details underlying this function, with Dicer as a key factor in the process.”

We have also edited the following sentence: “Thus, the mechanism by which Dicer regulates transcription, even in the absence of RdRP and small RNAs, may be conserved from fission yeast to mammals”.

Reviewer #2 (Remarks to the Author):

The authors' efforts to address my comments are appreciated, both in providing additional experiments and textual changes in the manuscript. I accept the authors' responses, with one exception. I appreciate the authors attempts to directly regulate major satellite transcription and while I agree with their interpretation that the data is highly suggestive of a function for major satellite transcription in Dicer mutant phenotypes, this is not directly demonstrated. For the most part the text is consistent with this interpretation. However in the discussion, they argue: 'In ES cells, we have shown that the Dicer1^{-/-} viability defect is due to transcription of the centromeric satellite repeats'. I think this sentence needs to be toned down before publication, in keeping with the rest of the revised manuscript.

We thank the reviewer for their comments and suggestions. We have edited the sentence in the discussion to reflect this, as follows:

"In ES cells, our data suggests that the viability defect of Dicer1^{-/-} cells is a consequence of transcription of the centromeric satellite repeats".

Reviewer #3 (Remarks to the Author):

In the revised version the authors have addressed all my concerns.

We thank the reviewer for their comments and suggestions.